**Interpreting cooling dates and histories from laser ablation in-situ (U-Th-Sm)/He thermochronometry: A modelling perspective.**

Christoph Glotzbach[1], Todd A. Ehlers [2,1]

[1.] Department of Geosciences, University of Tuebingen, Tuebingen, 72076, Germany

[2.] School of Geographical and Earth Sciences, University of Glasgow, G12 8QQ, UK

10      Corresponding author: christoph.glotzbach@uni-tuebingen.de

**Abstract**

Recent applications of the in-situ (U-Th-Sm)/He thermochronometry technique demonstrate its potential to address some of the analytical challenges associated with the whole-grain technique. In this study, we adapted state-of-the-art apatite and zircon production-ejection-diffusion models for application to in-situ dating methods, aiming to enhance the applicability of this technique to a broad range of geologic samples and applications. Our modifications to thermal history models include accommodation of the full range of stopping distances for alpha particles and cylindrical grain geometries. This investigation focuses on several key aspects of in-situ data interpretation: (i) exploring the relationship between in-situ dates and the position of ablation spots across individual grains, (ii) assessing differences and similarities between whole-grain and in-situ dates, (iii) determining optimal strategies and performance for reconstructing cooling histories from in-situ (U-Th-Sm)/He data, and (iv) reporting the effects of radionuclide zoning on (U-Th-Sm)/He thermochronology. Results indicate that the measured in-situ helium distribution is a function of grain size, ablation spot position and size, and cooling history. Together, these analytical and natural factors result in systematic variations in in-situ dates with distance from the grain rim. Therefore, similar to whole-grain analyses, robust interpretation requires determining grain geometry and the distance of the laser spot to the nearest prismatic face. In most cases, resulting in-situ dates are approximately 30% older than corresponding alpha-ejection corrected whole-grain dates, irrespective of the cooling rate and grain size. Whole-grain and in-situ dates are similar solely

for gem-size samples or samples exhibiting negligible diffusional helium loss, and thus spent more at surface temperatures compared to their transit time through the partial retention zone. Reconstruction of cooling histories using in-situ (U-Th-Sm)/He data can be achieved through single measurements in several grains with varying grain size and/or effective uranium content, or within a single grain with measurements taken at different distances from the

grain rim. In addition, statistical analysis of a large compilation of measured radionuclide variations in apatite and zircon grains reveals that radionuclide zoning strongly impacts whole-grain analyses, but can be directly measured with the in-situ method. Overall, our results suggest that in-situ measurements for (U-Th-Sm)/He date determination offer a means to extract meaningful cooling signals from samples with poor reproducibility from traditional

whole-grain techniques.

### 1.0 Introduction


Alpha decay of radiogenic isotopes and related ingrowth of $^4$He in crystal grains is the basis of the widely applicable (U-Th-Sm)/He method (e.g. Lippolt et al., 1994, Wolf et al., 1996, Farley, 2002). A wide variety of minerals incorporate trace amounts of naturally occurring alpha-emitting isotopes such as U, Th, and Sm. Among those minerals, apatite and zircon

have some favourable properties, making them a common choice for a wide range of applications to problems in tectonics and surface processes (e.g., Farley, 2000, 2002; Gallagher et al., 1998; Reiners and Ehlers, 2005; Malusà and Fitzgerald, 2019). Most importantly, apatite and zircon are abundant in many rock types, have a well-defined He diffusion behaviour (e.g., Farley, 2000; Reiners, 2005; Hourigan et al., 2005; Flowers et al.,

2009; Guenthner et al., 2013), and are sensitive to upper crustal temperatures (e.g., Ehlers, 2005; Reiners and Brandon, 2006). Most applications of apatite and zircon (U-Th-Sm)/He thermochronometry make use of this and invert (U-Th-Sm)/He data to retrieve cooling histories of exhumed rocks (e.g. Wolf et al., 1996). The majority of (U-Th-Sm)/He thermochronometry studies use multiple whole-grain measurements from a single sample,

often in combination with other thermochronometric data (e.g. Flowers 2009; Guenthner et al., 2017; Falkowski et al., 2023). This is possible because He diffusion in apatite and zircon is controlled by grain size and accumulated radiation damage, both of which vary from grain to grain and thus lead to sample- and thermal history-specific relationships between these

parameters. An alternative method to reveal the near-surface thermal history of rocks is the
$^4$He/$^3$He method (Shuster and Farley, 2004), which indirectly measures the He profile by
stepwise degassing of He from proton-irradiated apatite grains.

Irrespective of the method applied, deriving accurate cooling histories is often difficult
because of biases introduced by (i) fluid inclusion or inclusion of radionuclide-rich mineral
phases (e.g., Farley, 2002; Ehlers and Farley, 2003; Vermeesch et al., 2007; Danišík et al.,
2017), (ii) implantation of He from radionuclide-rich phases from outside the grain (Spiegel
et al., 2009), and (iii) radionuclide zonation and related variability of diffusion caused by
radiation damage (e.g., Hourigan et al., 2005; Fox et al., 2014, Anderson et al., 2017). Careful
selection of euhedral grains free of visible inclusion can prevent large biases caused by the
first process. In the case of detrital studies where understanding the date distribution is the
objective, excluding grains can introduce bias in the resulting date distributions. In-situ (U-
Th-Sm)/He method theoretically provides less biased results since unsuitable parts of grains
can be excluded from analyses (e.g., Tripathy-Lang et al., 2013). Radionuclide zoning and
the implantation of He are usually not accounted for in common (U-Th-Sm)/He protocols.
Implanted radiation damage in apatite and zircon and zonation, especially in zircon, increase
the variance in whole-grain (U-Th-Sm)/He dates and are likely the main causes for
overdispersed dates (e.g. Flowers et al., 2009; Horne et al., 2016, 2019).

In this regard, the introduction of the in-situ (U-Th-Sm)/He method by Boyce et al. (2006),
has the potential to resolve some of the issues related to whole-grain analyses. However, in-
situ dating has not become a routine alternative to whole-grain measurements, despite several
studies demonstrating the reliability of dating large and/or rapidly cooled monazite, zircon,
and apatite age standards (e.g. Boyce et al., 2006; Tripathy-Lang et al., 2013; Evans et al.,
2015). One potential issue is the complex geometric relation between radionuclides and
produced He, originating from long-alpha stopping distances (up to several tens of microns)
and separation of daughter product from sourced parental radionuclide (e.g., Farley et al.,
1996). Another potential issue is that more common small grains with less rapid cooling
suffer from partial He loss by diffusion and thus should result in older whole-grain dates
compared to in-situ (U-Th-Sm)/He dates (e.g. Tripathy-Lang et al., 2013). He loss by
diffusion mainly occurs in the outer part of a grain (Fig. 1A). An in-situ He measurement in
the center of a grain (Fig. 1B), results in a date that is similar to a whole-grain date only for
cooling scenario 1 that involves rapid cooling to the surface, followed by a prolonged stay at
the surface (Fig. 1C). In cooling scenarios 2 and 3 that involves a longer time at temperatures

where He diffusion is occurring, in-situ dates are older compared to whole-grain dates (Fig. 1C).

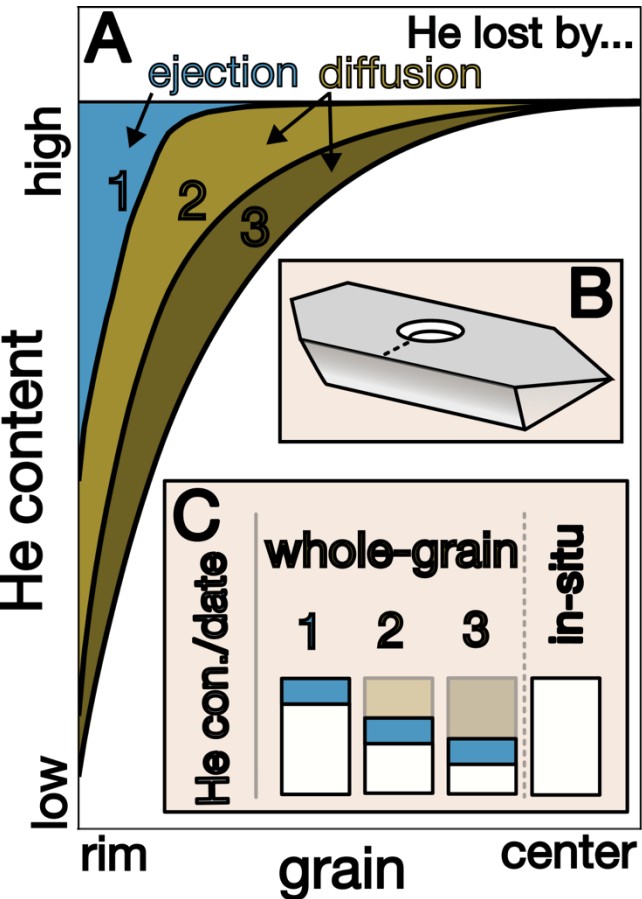

Fig. 1: Whole-grain vs. in-situ (U-Th-Sm)/He dates for end-member He profiles. A) Modelled He profiles for three cooling histories assuming uniform radionuclide distributions: 1) Rapid cooling to surface temperature, followed by a prolonged stay at the surface. 2) Constant slow cooling. 3) Prolonged stay in the partial retention zone, followed by rapid cooling to the surface. The blue area is the He content lost by alpha ejection, whereas He lost
by diffusion is shown in brown. B) Cylindrical grain with ablation pit and location of He profiles in A. C) Corresponding He concentrations and resulting whole-grain and in-situ (U-Th-Sm)/He dates for the cooling histories in 1, 2 and 3. Note that ejected He (blue bar) is added to the measured He (white bar) in the whole-grain approach based on grain geometry (Ft-correction).  The corresponding in-situ dates calculated for a central pit are identical
irrespective of cooling histories and are similar to the whole-grain date only for the rapid cooling scenario (1). In all other cases, He is lost by diffusion, especially in the outer part of grains and in-situ (U-Th-Sm)/He dates are older compared to a whole-grain (U-Th-Sm)/He dates.

In this study, we explore the theoretical measurement procedures required to interpret in-situ
        (U-Th-Sm)/He dates to retrieve cooling histories from multiple measurements in several
        grains or from a single grain. To do this, we simulate the He concentration across grains as a
        function of grain size/shape, radionuclide zoning and cooling history. These predicted He-
        distributions across grains are used to investigate the theoretical relationship between the size
and position of in-situ laser ablation spots and the corresponding in-situ (U-Th-Sm)/He dates.
        The in-situ modelled dates are then compared to modelled whole-grain dates to identify the
        usability and limitations of each technique. In addition, the effect of radionuclide zoning in
        apatite and zircon on whole-grain dates is studied based on a large LA-ICP-MS dataset. We
        find that theoretically single in-situ (U-Th-Sm)/He measurements from different grains from
the same sample, or multiple measurements within a single grain can be successfully inverted
        to retrieve consistently complex cooling histories similar to whole-grain analyses.

        **2.0 Methods**


                    2.1 Modelling approach for He production, ejection and diffusion

        The in-situ (U-Th-Sm)/He method is based on the extraction of He, U, Th and Sm from a
        small fraction of the grain using a laser ablation system (e.g. Boyce et al., 2006; Tripathy-
Lang et al., 2013; Evans et al., 2015; Anderson et al., 2017; Pickering et al., 2020). Ablation
        pits can have a radius of a few tens of μm and depths of a few μm (e.g., from an excimer
        laser). Importantly, the ratio of He to U, Th and Sm (and therefore the date) varies with the
        size and position of the laser ablation measurements and likely differs from corresponding
        whole-grain (U-Th-Sm)/He dates. This, however, does not mean that dates are wrong or not
interpretable; instead, they require a refinement of the interpretation steps commonly applied
        to (U-Th-Sm)/He data.
        Whole-grain (U-Th-Sm)/He analyses often use a sphere-equivalent radius and assume
        spherical isotropic diffusion to estimate whole-grain He production, ejection, and diffusion in
        apatite/zircon crystals (e.g. Farley et al., 1996; Meesters and Dunai, 2002). More effort is
required to match grain geometry for the in-situ (U-Th-Sm)/He method since long-alpha
        stopping distances (up to several tens of μm) result in a complex geometric relation between
        the location of radionuclides (U, Th and Sm) and the resulting position of produced He. Most
        apatite grains have a prismatic geometry, with typical length/radius ratios of 4-8 (Farley,

2000). Loss of He by ejection and diffusion mostly occurs perpendicular to the crystallographic c-axis in prismatic grains such as apatite and zircon, and thus the He profile should be approximated for most grains with a finite cylinder model (Meesters and Dunai, 2002). Farley et al. (2011) provide a method to transform measured element concentrations from cylindrical grains into an equivalent spherical-geometry, thereby providing input in the commonly used modelling software HeFTy (e.g. Danišík et al., 2017). Complementary to this approach, here we used the available spherical model implemented in HeFTy (Ketcham, 2005) and modified it to handle an infinite cylinder geometry. The latter should be a good approximation for in-situ measurements outside the tips/caps of the analyzed grains where alpha-ejection effects become more significant. The advantage of an infinite cylinder model (compared to a finite cylinder model) is that it can be solved in 1D and thus runs as fast as the spherical model, a prerequisite for applying efficient inverse thermal history modelling. We adjusted the available He production, ejection, and diffusion models implemented in HeFTy (Ketcham, 2005; Flowers et al., 2009; Guenthner et al., 2013) to handle an infinite cylinder geometry. More specifically, we implemented our changes to the existing C++ code (kindly provided by R. Ketcham) that simulates He diffusion following the RDAAM (apatite, Flowers et al., 2009) and ZRDAAM (zircon, Guenthner et al., 2013) diffusion and annealing models. The modified version of RDAAM and ZRDAAM code is available from the Zenodo repository (**https://zenodo.org/records/10531763**).

### 2.2 Geometric considerations for He production, ejection and diffusion

The amount of He produced vs. ejected and diffused out of the grain depends on the concentration and distribution of parent isotopes and the grain morphology. These effects differ in spherical and cylindrical grains, especially if grains are zoned. Spherical zonation has been implemented in diffusion models for spheres (e.g., in HeFTy), which we also explore here for an infinite cylinder geometry. For simplicity, we assume that radionuclide zoning is symmetric around the c-axis for cylindrical grains (Fig. 2). Note that this might not always be applicable, especially to zircon grains, which apart from concentric parent nuclide distributions also reveal patchy/chaotic patterns (e.g. Chew et al. 2017; Danišík et al. 2017; Fox et al. 2017). It is therefore recommended that such a simplified approach only be applied to grains satisfying concentric parent nuclide distributions. Radionuclide zoning and grain size (especially the distance to the grain rim) control the amount of He along the radius (r-

axis), without diffusion. The He distribution along the r-axis is derived by calculating the intersecting lines of all alpha-ejection spheres (ranging from ~6 to ~40 µm) and internal cylinders with a radius defined by the grain size and grid spacing. The intersection line can consist of two closed curves, a continuous line, or, if the cylinder and the sphere are tangential to each other at one point, the line forms an 'eight' geometry, also known as Viviani's curve (Fig. 2A).

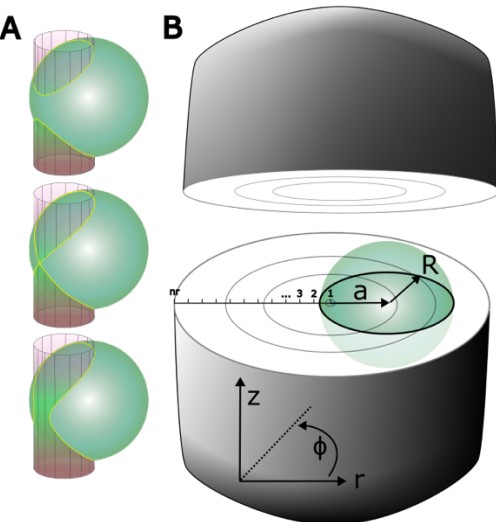

Fig. 2: Geometric relationship between alpha-ejection spheres (green) and intersecting inner grain coaxial cylindrical surface representing variable radionuclide concentrations in a cylindrical grain (light grey lines). A) The length of a line defining the intersection between the cylinder and a sphere depends on the size of each object and its position. B) Assumed cylindrical grain with radionuclide zoning parallel to the z-axis is intersected by an alpha-ejection sphere with radius R and distance from the centre of a. The modelled He profile is discretized from the centre of the grain to the rim with nodes from $i=1...nr$.

The procedure for calculating the amount of He in an infinite cylinder without diffusion along the r-axis is:

1. The grain is discretized by a number of cylinders $(r,\Phi,z)$, and the circular shape of the cylinders in $r,\Phi$-plane is transformed in $x,y$-coordinates:

$$x_{r,\Phi} = rcos(\Phi) \tag{1}$$
$$y_{r,\Phi} = rsin(\Phi) \tag{2}$$

2. The $z$-coordinates of the intersection line between the cylinder and the alpha-ejection sphere (the yellow line in Fig. 2A) are calculated with:

$$z_{r,\Phi,R,a} = R^2 - \left(x_{r,\Phi} - a\right)^2 - y_{r,\Phi}{}^2 \tag{3}$$

where $R$ is the radius of the alpha-ejection sphere, and $a$ is the distance between the centre of the cylinder and the alpha-ejection sphere, and Phi and r are the same as in Fig. 2B.

3. The length of the intersection line is calculated with the Pythagorean theorem:

$$l_{r,R,a} = 2 \sum_{i=1}^{n\Phi-1} \sqrt{ \begin{array}{c} \left(x_{r,R,a}(\Phi_i) - x_{r,R,a}(\Phi_{i+1})\right)^2 + \\ \left(y_{r,R,a}(\Phi_i) - y_{r,R,a}(\Phi_{i+1})\right)^2 + \left(z_{r,R,a}(\Phi_i) - z_{r,R,a}(\Phi_{i+1})\right)^2 \end{array} } \tag{4}$$

where $\Phi$ has been discretized from 0 to $2\pi$ into $i=1...n\Phi$.

4. Next, the length is normalized to unity:

$$\check{l}_{r,R,a} = \frac{l_{r,R,a}}{\sum_{i=1}^{na} l_{r,R}(a_i)} \tag{5}$$

where $a$ has been discretized from r=0 to the rim into $i=1...na$.

5. Finally, we derived the radionuclide-specific concentration ($C_{I,a}$) for isotopes ($I$) and points

($a$) along the $r$-axis with:

$$C_{I,a} = \sum_{j=1}^{nR} F_{I,j} \sum_{i=1}^{nr} \check{l}_{r,a}(R_j) C(r_i) \tag{6}$$

where $F_{I,j}$ is the fractional contribution of an isotope-specific stopping distance and $C$

is the radionuclide concentration depending on $r$.

The resulting He distribution is very similar to a spherical grain but with an overall higher concentration (for similar radii) since we assume an infinite length of the cylinder (Fig.

3A,B). Consequently whole-grain and in-situ (U-Th-Sm)/He dates of a cylindrical grain are significantly older (in-situ: 19-27%, whole-grain: 11-12%) compared to a spherical grain with similar radii (Fig. 3C,D). Incorporating He diffusion and alpha-stopping distances leads to smooth (uniform radionuclides) or complex (zoned grains) He profiles (Fig. 3,6).

## 2.3 Calculation of He diffusion

Assuming a spherical grain geometry provides a good estimate of whole-grain He diffusion in apatite crystals (e.g., Farley et al., 1996; Meesters and Dunai, 2002). However, most apatite and zircon grains have a prismatic shape with hexagonal (apatite) and quadratic (zircon) cross-sections. Efficient modelling of He profiles requires a 1D solution of the diffusion equation and therefore a round cross-section, which can accurately predict He concentrations in apatite and zircon (cf. Eq. 19 and 20 and section 2.5). In the following, we solved the production and diffusion equation for an infinite cylinder (Farley, 2000). The 3D diffusion equation in a cylinder is:

$$\frac{1}{r}\frac{\partial}{\partial r}\left(rK\frac{\partial v}{\partial r}\right) + \frac{1}{r^2}\frac{\partial}{\partial \phi}\left(K\frac{\partial v}{\partial \phi}\right) + \frac{\partial}{\partial z}\left(K\frac{\partial v}{\partial z}\right) + A_0 = \frac{\partial v}{\partial t} \tag{7}$$

where $v$ is the He quantity, $K$ is the diffusivity, $t$ is time, $A_0$ is the volumetric He production, and $r, z$ and $\phi$ are the radial, vertical and azimuth positions (e.g., Fig. 2B). Assuming an infinite length of the cylinder and that He does not vary with $z$ and $\phi$, the equation (Eq. 7) simplifies to:

$$\frac{1}{r}\frac{\partial}{\partial r}\left(rK\frac{\partial v}{\partial r}\right) + A_0 = \frac{\partial v}{\partial t} \tag{8}$$

Using the product rule, we get:

$$\frac{K}{r}\frac{\partial v}{\partial r} + K\frac{\partial^2 v}{\partial r^2} + A_0 = \frac{\partial v}{\partial t} \tag{9}$$

We solved Equation 9 with an implicit Euler finite difference method with the following assumptions: (i) grain symmetry (including geometry and radionuclide distribution) around

the *z*-axis, (ii) zero-flux Neumann boundary condition in the centre of the grain (Eq. 10), and (iii) zero He concentration at the grain boundary (Eq. 11):

$$\frac{\partial v}{\partial r} = 0 \text{ for } r=0 \tag{10}$$

$$v = 0 \text{ for } r=rim \tag{11}$$

Reformulating equation (Eq. 9 and 10) with the implicit Euler method yields:

$$\frac{v_i^{t+1}-v_i^t}{\Delta t} = \frac{K}{r}\frac{v_{i-1}^{t+1}-v_{i+1}^{t+1}}{\Delta r} + K\frac{v_{i-1}^{t+1}-2v_i^{t+1}+v_{i+1}^{t+1}}{\Delta r^2} + A_0 \qquad \text{for r>0 \& r<rim} \tag{12}$$

$$\frac{v_i^{t+1}-v_i^t}{\Delta t} = K\frac{v_{i-1}^{t+1}-v_{i+1}^{t+1}}{\Delta r} + A_0 \quad \text{for r=0} \tag{13}$$

Since *i*=-1 is not defined, but similar to *i*=+1, it is common to instead use the second derivative and Eq. 13 changes to:

$$\frac{v_i^{t+1}-v_i^t}{\Delta t} = K\frac{2v_{i+1}^{t+1}-2v_i^{t+1}}{\Delta r^2} + A_0 \text{ for i=0} \tag{14}$$

Solving Equation 14 requires a tridiagonal matrix whereby all unknows (t+1) are brought to the left-hand side:

$$(1 + 2D)v_i^{t+1} - 2Dv_{i+1}^{t+1} = v_i^t + A_0\Delta t \qquad \text{for i=0} \tag{15}$$

$$\left(-\frac{D\Delta r}{r} - D\right)v_{i-1}^{t+1} + (1 + 2D)v_i^{t+1} + \left(\frac{D\Delta r}{r} - D\right)v_{i+1}^{t+1} = v_i^t + A_0\Delta t \quad \text{for r>0 \& r<rim} \tag{16}$$

$$v_i^{t+1} = 0 \qquad \text{for } r=rim \tag{17}$$

where *D* is $K\Delta t/\Delta r^2$, and the corresponding tridiagonal matrix needed to solve for diffusion in an infinite cylinder is given by:

$$\begin{pmatrix} 1+2D & -2D & 0 & 0 & \dots & 0 \\ -\frac{D\Delta r}{r} - D & 1+2D & \frac{D\Delta r}{r} - D & 0 & \dots & 0 \\ 0 & -\frac{D\Delta r}{r} - D & 1+2D & \frac{D\Delta r}{r} - D & \dots & 0 \\ \vdots & \vdots & \vdots & \vdots & \ddots & \vdots \\ 0 & 0 & 0 & 0 & 0 & 1 \end{pmatrix} \begin{pmatrix} v_{i=0}^{t+1} \\ v_{i=1}^{t+1} \\ v_{i=2}^{t+1} \\ \vdots \\ v_{i=r+1}^{t+1} \end{pmatrix} = \begin{pmatrix} v_{i=0}^t + A_0\Delta t \\ v_{i=1}^t + A_0\Delta t \\ v_{i=2}^t + A_0\Delta t \\ \vdots \\ 0 \end{pmatrix} \tag{18}$$

The resulting He profiles for infinite cylinders have similar shapes as a sphere, but higher He
      concentrations than a sphere with the same radius (Fig. 2A,B). The difference in
      concentration between the infinite cylinder and sphere geometry (for constant cooling) is in
      the range of 15-20% in the centre of the grain, comparable to previous observations
      (Meesters and Dunai, 2002). Corresponding in-situ dates are 19-27% older in an infinite

cylinder compared to a sphere model with a similar grain radius, while whole-grain dates (not
      shown) differ by 11-12%. The choice of geometry to model in-situ (U-Th-Sm)/He dates
      matters even more than for the whole-grain method.

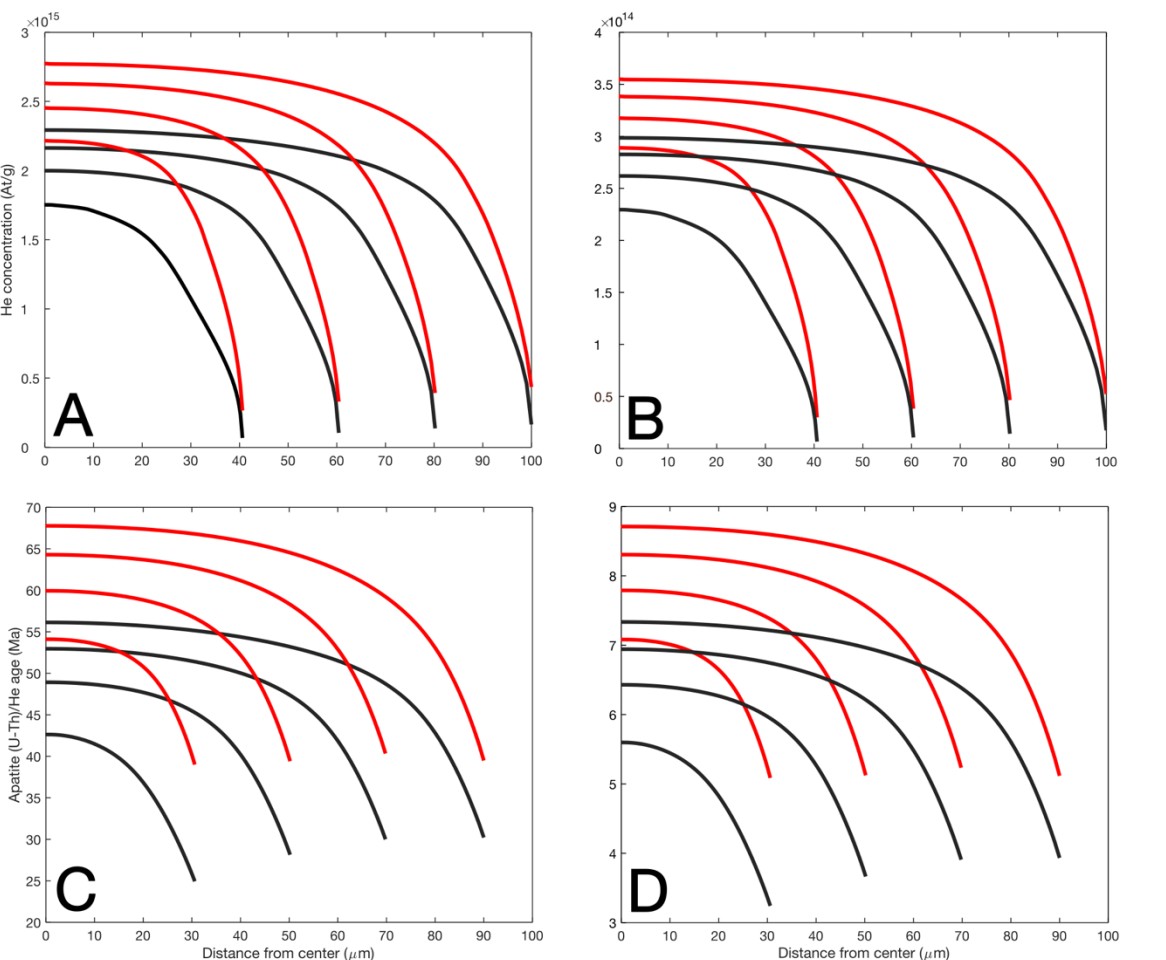

Fig. 3: Difference between He diffusion profiles and in-situ (U-Th-Sm)/He dates in a sphere

(black) and infinite cylinder (red) perpendicular to the c-axis with grain radii of 40, 60, 80
      and 100 μm. All profiles are calculated for apatite grains applying the production, ejection
      and diffusion with homogenous U, Th and Sm distributions (10 ppm), and constant cooling
      rates of 1°C/Myr (A,C) and 10°C/Myr (B,D). In-situ (U-Th-Sm)/He dates of infinite cylinder

grains are 19-27% older compared to spheres with similar radii, whereas corresponding

whole-grain (U-Th-Sm)/He dates differ by 11-12%.

A cylindrical model is a good approximation for the He profile in hexagonal apatite grains (Meesters and Dunai, 2002), but it is unclear what radius should be used to estimate the He profile. To determine the appropriate cylinder radius to approximate diffusion in a hexagonal

grain, we calculated the 2D (cross-sectional) He distribution of an infinitely long symmetrical hexagonal grain with circumradius $r_c$ between 30-50 μm and corresponding infinite cylinders with variable radii. We calculated the difference between the mean He profile of the hexagonal and cylindrical grains and found that the circle-equivalent radius (CER) of a symmetrical hexagonal grain is simply the radius of a circle with a similar area:


$$CER_{ap} = \sqrt{\frac{\frac{3\sqrt{3}}{2}r^2}{\pi}} \approx 0.9094r \qquad (19)$$

where $r$ is the outer radius (touching all vertices) of a symmetrical hexagon. Equivalent to a zircon with a quadratic cross-section, we derived the following:


$$CER_{zr} = \sqrt{\frac{2r^2}{\pi}} \approx 0.5642r \qquad (20)$$

where $r$ is the outer radius of a quadrate.


### 2.4 Implementation of alpha-stopping distances

During alpha decay, energy is released that leads to long alpha-stopping distances (e.g.,

Bragg and Kleeman, 1905). The common radiogenic isotopes $^{238}$U, $^{235}$U, $^{232}$Th, and $^{147}$Sm release alpha decay energies between 2233 keV ($^{147}$Sm to $^{143}$Nd) to 8784 keV ($^{212}$Po to $^{208}$Pb as part of the decay chain of $^{232}$Th). In total, the alpha decay of these radionuclides produces 216 different energies, each occurring with a different probability (Fig. 4). The relation between energy and stopping distance has been measured and calculated and is easily

accessible from the SRIM data collection (e.g., Farley et al., 1996; Ketcham et al., 2011). Alpha particles produced from [238]U, [235]U, and [232]Th have stopping distances between ~11 and ~40 μm in apatite (Fig. 4A-C), and those derived from [147]Sm have a single stopping distance of ~6 μm. Integration of the stopping distance distribution yields the average stopping distances (Fig. 4A-C), commonly used to approximate He distribution profiles and $F_T$

correction factors for whole-grain analyses (e.g., Ketcham et al., 2011). Note that the reported values using the SRIM 2013 data show only very small differences from those of Ketcham et al. (2011), which is based on SRIM 2008.

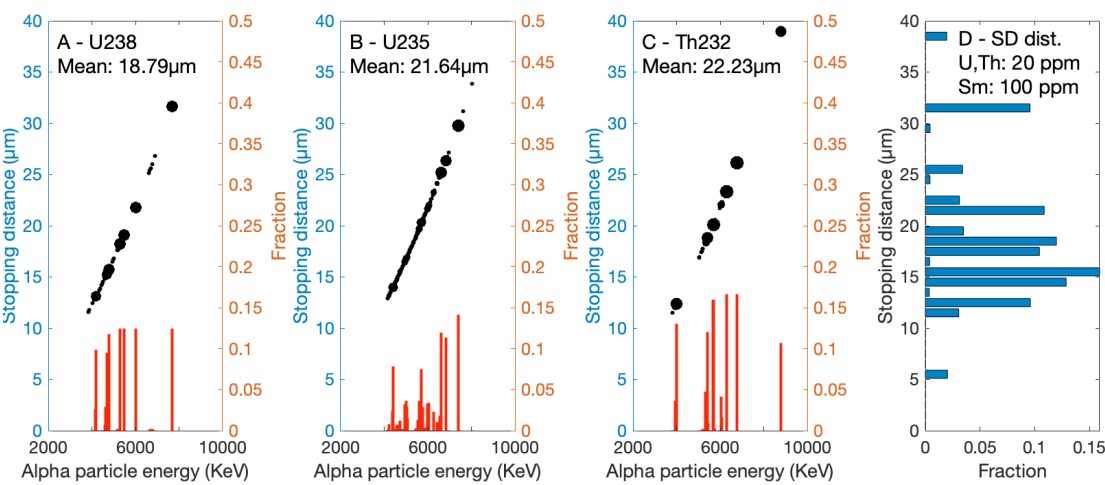

Fig. 4: Alpha particle energy spectra of [238]U (A), [235]U (B), and [232]Th (C) and the corresponding stopping distance spectra and mean stopping distances derived from SRIM2013 data assuming a fluorine apatite with a density of 3.2 g/cm$^3$. The stopping distance distribution (SD dist.) of a typical apatite with 20 ppm U and Th, and 100 ppm Sm is shown in (D).


Depending on the relative concentration of radionuclides, each mineral crystal will have a grain-specific alpha-stopping distance distribution (Fig. 4D). The majority of stopping distances in apatite are between 11 and 26 μm, with additional peaks at 6 μm, 32 μm, and 39 μm for a common apatite (Fig. 4D). Stopping distances in zircons are shorter and less

variable, ranging from 9 to 32 μm, while the majority are between 10 and 26 μm long (Fig. S1). Due to the long alpha-stopping distances, the in-situ measured He in an infinitely small area within the grain is produced from the surrounding ~6 to 39 μm and ~9 to 32 μm in apatite and zircon, respectively. Importantly, He originates from the surface of spheres with a radius corresponding to the stopping distance distribution (Fig. 2,5). This does not have large

consequences for grains with a homogenous radionuclide distribution, but He and

radionuclide distributions do not follow a 1:1 relation in the case of radionuclide

heterogeneity in grains (Fig. 5).

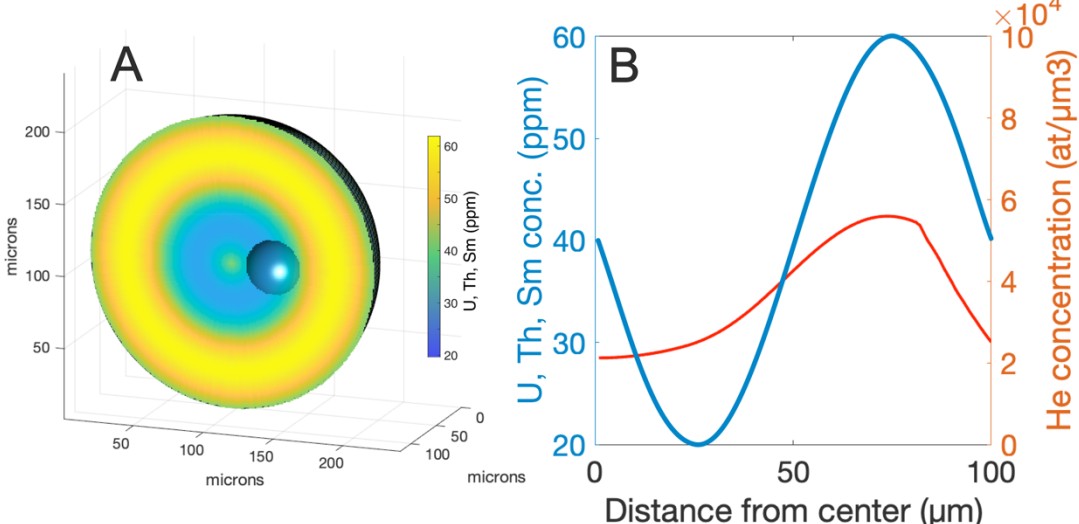

Fig. 5: Relationship between radionuclide zonation and resulting He profiles in a spherical

apatite grain. A) Spherical grain with radial U, Th and Sm concentrations between 20 to 60

ppm, respectively. The small half-sphere corresponds to a stopping distance of 20 μm. B) U,

Th, Sm and resulting He concentrations from the core to the rim of the grain shown in A.

As a consequence, we adjusted the original RDAAM and ZRDAAM c++ implementation of

HeFTy to (i) handle the full spectrum of stopping distances (instead of using an averaged

value) of respective radionuclides and (ii) incorporate inner grain radionuclide variations.

We tested our extended implementation against the original implementation for a theoretical

spherical apatite grain (Fig. 6). The resulting whole-grain dates are indistinguishable from

each other but the He profiles produced are smoother and, in some cases, show distinct

differences (Fig. 6). Considering the full spectrum of stopping distances results in an overall

lowering of the He concentration when approaching the grain rim for uniform radionuclide

distributions (Fig. 6A). The incorporation of longer stopping distances (up to 39 μm) results

in reduced He production at a distance between the longest stopping distance and the mean

stopping distance from the grain rim. The opposite effect (higher He concentrations nearer to

the grain rim) originates from stopping distances shorter than the mean stopping distance

(Fig. 6A). Spherical grains with a grain radius smaller than the longest stopping distance (39

μm) but larger than the mean stopping distance (~20 μm) show lower He concentrations since

the production in the grain core is zero when the grain radius is smaller than the stopping

distance (Fig. 6A). Variations in radionuclides, such as at the boundary between the grain and exterior require consideration of the full spectrum of stopping distances for the case of in-situ (U-Th-Sm)/He analyses close to the grain rim (within 39 μm from the grain rim). Similarly, a mean stopping distance approach results in dissimilar He profiles in zoned grains with radionuclide variations compared to considering the full spectrum (Fig. 6B). To give an example, a two times higher radionuclide concentration between half and three-quarters of the radius (measured from the centre) results in the largest differences in the He concentration in the centre of the grain. The latter is usually the target of in-situ (U-Th-Sm)/He analyses. We suggest that any in-situ (U-Th-Sm)/He analyses require considering the full spectrum of significantly contributing alpha-stopping distances. Although not investigated here, the $^4$He/$^3$He method might also benefit from considering this to predict more accurate He profiles (e.g., Shuster and Farley, 2004).

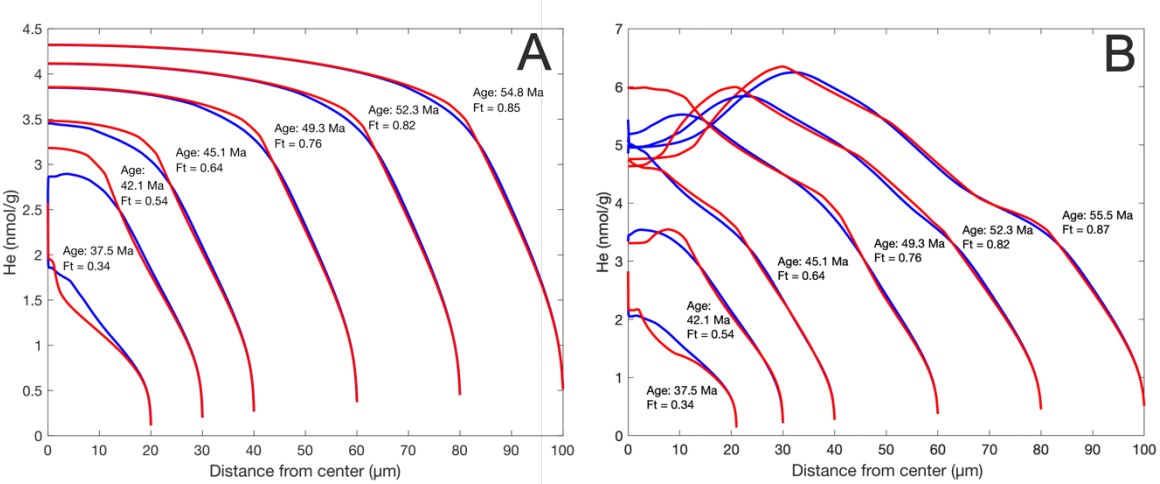

Fig. 6: Apatite He profiles and (U-Th-Sm)/He dates for a cooling rate of 1°C/Myr, variable grain sizes, and mean (red, original implementation) or complete stopping distances (blue, our implementation). A) Uniform U, Th and Sm concentration of 10 ppm. B) Same as in A) but with two times higher radionuclides between the half radius and ¾ radius.

2.5 Circle Equivalent Radius (CER)

Modelling (U-Th-Sm)/He data is usually accomplished using a transient 1D axial-symmetric parameterization, where different grain morphologies are approximated with a spherical geometry with similar volume-to-surface area ratios. With this approach, a solution is calculated in 1-D (as a function of radius), and then integrated over the volume of the

equivalent sphere. It has been shown that this approach is accurate for common grain
morphologies within a few percent, whereas an infinite cylinder can have a deviation of up to
7% (e.g. Meesters and Dunai, 2002). Here we explore if such an approach also applies to
accurately estimating He profiles within grains. We modelled the He distribution of an
infinite symmetrical hexagonal prism with an outer radius of 50 μm (inner radius of 43.3
μm). We compared these results to those calculated with an infinite cylinder (Fig. 7). The
CER for such a grain is ~45.5 μm. The mean He profile of the hexagonal prism after
averaging all possible profiles from the centre of the grain to the edge are nearly identical to a
cylinder with a radius of 45.5 μm. This result is irrespective of the cooling history (Fig. 7).
The He profile deviates substantially at the outermost 5-10 μm between the long and short
axis of a hexagon, which should be discarded if it is not exactly known where the short and
long axes are relative to the location of measurement.

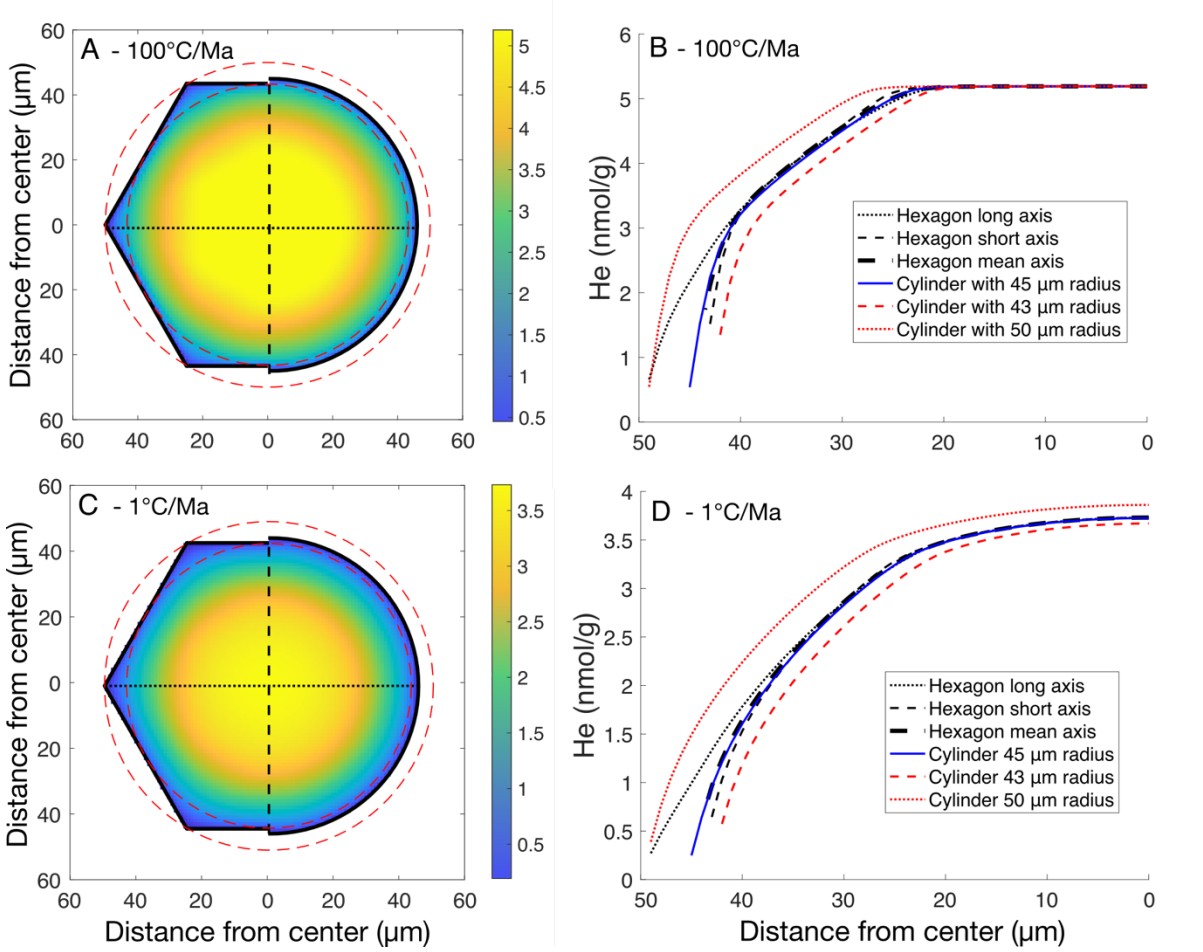

Fig. 7: He concentration in infinite hexagonal prism and cylinder. A) He concentration map
modelled with RDAAM for a symmetrical hexagonal prism with an outer radius of 50 μm
and cylinder with a radius of 45.5 μm, and 10 ppm U, Th and Sm concentrations, and rapid
cooling (100°C/Myr) to surface temperature (10°C) at 60 Ma. B) Corresponding He profiles

for a hexagonal prism and infinite cylinder with different radii. C/D) Similar to A/B but for a

constant cooling rate of 1°C/Myr.

2.6  Analytical uncertainty

In-situ (U-Th-Sm)/He dating relies on measuring pits that are only a few tens of microns in

size, leading to increased analytical uncertainty, particularly in He measurements, compared

to conventional whole-grain analyses. A typical cylindrical grain used in whole-grain

measurements, with a diameter of 100 μm and a length of 200 μm, has a volume

approximately 1000 times greater than a standard in-situ pit with a diameter of 30 μm and 5

μm depth. Consequently, in-situ analyses face significant analytical uncertainties, especially

when applied to young samples or grains with low radionuclide concentrations. To report

these limitations, we used the detection limit determined in our laboratory at the University of

Tuebingen to model analytical uncertainties. The standard deviation of repeated line blanks

($SD_{lb}$) gives a $^4$He of 0.000079 ncc or $2.11 \times 10^6$ atoms. This allows for estimating the

analytical uncertainty for modelled in-situ He content ($^4$He$_m$) using the following equation:

$$u = \sqrt{\left(\frac{SD_{lb}}{{}^4He_m}\right)^2} \qquad\qquad \text{Eq. 21}$$

This equation does not account for uncertainties related to the required measurement of

radionuclides, which are generally small and around 2%.

**3.0  Results**

3.1  Whole-grain vs. in-situ (U-Th-Sm)/He dates

Whole-grain (U-Th-Sm)/He dates reflect the production, ejection, diffusion, and alpha-

ejection correction for the complete grain. In contrast, in-situ (U-Th-Sm)/He dates, if

measured in the centre of grains, are not affected by alpha ejection, less affected by diffusion,

and do not require an alpha-ejection correction (e.g., Tripathy-Lang et al., 2013).

Theoretically, in-situ dates will, in most cases, differ from whole-grain dates from similar

grains. The majority of in-situ (U-Th-Sm)/He studies applied so far used large crystals with

homogenous radionuclide distributions and/or rapidly cooled samples to enable comparison of the results to whole-grain measurements (e.g., Boyce et al., 2006; Horne et al., 2016).

Making this method applicable to small and/or slowly cooled grains requires understanding the relationship between the grain size, position and size of the ablation spots, radionuclide distribution, and resulting (U-Th-Sm)/He dates. To investigate these effects we considered several scenarios.

First, we modelled whole-grain and in-situ (U-Th-Sm)/He dates as a function of cooling rate

(1, 10 and 40°C/Myr) for radionuclide concentrations of 10 ppm and a grain radius of 100 μm (Fig. 8A-C). Modelled whole-grain dates are 49, 6.5 and 1.9 Ma for cooling rates of 1, 10 and 40°C/Myr, respectively. In-situ (U-Th-Sm)/He dates vary as a function of their measurement position in the grain. Assuming similar grain parameters and cooling rates, in-situ dates range between 48-65 Ma (1°C/Myr), 6.3-8.3 Ma (10°C/Myr) and 1.8-2.4 Ma (40°C/Myr) for a spot

diameter of 20 μm and grain radius of 100 μm (Fig. 8A-C). Modelled in-situ dates measured in the centre of grains are older than the whole-grain dates because the fraction of He lost by diffusion is smallest in the centre of grains and increases towards the grain boundary, as does He loss by alpha-ejection. Accordingly, in-situ dates become progressively younger towards the grain rim. A larger laser spot size averages over a larger area and may incorporate areas

affected by He loss. A larger spot size, therefore, leads to younger dates, and a smaller spot size can be expected to produce less variation in dates, especially when analyzing smaller grains. Modelled analytical uncertainties limit the applicability of the in-situ (U-Th-Sm)/He method for young grains (Fig. 8A,B). A spot diameter >50 μm and therefore a grain with a diameter of ~100 μm is required to reach uncertainties <100% (Fig. 8A,B).

Second, we simulated the effect of grain size on whole-grain and in-situ (U-Th-Sm)/He dates for a cooling rate of 1°C/Myr, radionuclide concentrations of 10 ppm and grain radii of 100, 80, 50 and 40 μm (Fig. 8C-F). Whole-grain dates decrease as a function of grain size from 49 to 39 Ma, while in-situ dates consistently result in older dates. In-situ measurements with similar spot diameter (e.g., 20 μm) sample larger fractions of areas affected by He loss and,

therefore, in-situ dates become less sensitive to the measurement position in the grain for smaller grains. In the most extreme case where the spot diameter corresponds to the grain radius the alpha-ejection corrected in-situ date would match the whole-grain date. In practice, the spot size also depends on the expected He concentration and must be determined based on the detection limit of He measurable with the instrumental setup and the maximum depth of

laser pits that can be measured accurately.

Third, we studied the in-situ date dependency for cases in which grains have not been ground and polished to the exact centre of the grain (Fig. 8G,H). In-situ dates become progressively younger towards the grain rim compared to a measurement in the centre of the grain (Fig. 8G,H). In a large grain (100 μm radius, Fig. 8G), dates are similar within <40 μm from the central plane of the grain. In-situ dates in smaller grains are more sensitive to the position of the measurement relative to the grain rim (Fig. 8H).

In summary, uniform cooling yields in-situ (U-Th-Sm)/He dates that are older than whole-grain dates, and dates strongly vary as a function of the measurement position relative to the grain rim. The results obtained here for apatite also apply to zircons, as has been revealed by modelling in-situ dates as a function of grain size, and position and size of the ablation spots with the ZRDAAM approach (Fig. S2). Measuring grain size and geometry, and the laser spot position relative to the grain rim is essential for correctly interpreting in-situ (U-Th-Sm)/He dates. The grain size and geometry, and location of laser pits on the grain surface can be easily determined with an optical microscope, whereas estimating the pit location in the vertical direction is difficult. A rough estimate (±5 μm) can be gained by focusing on the contact between grain and embedding media and measuring the distance to the exposed grain surface.

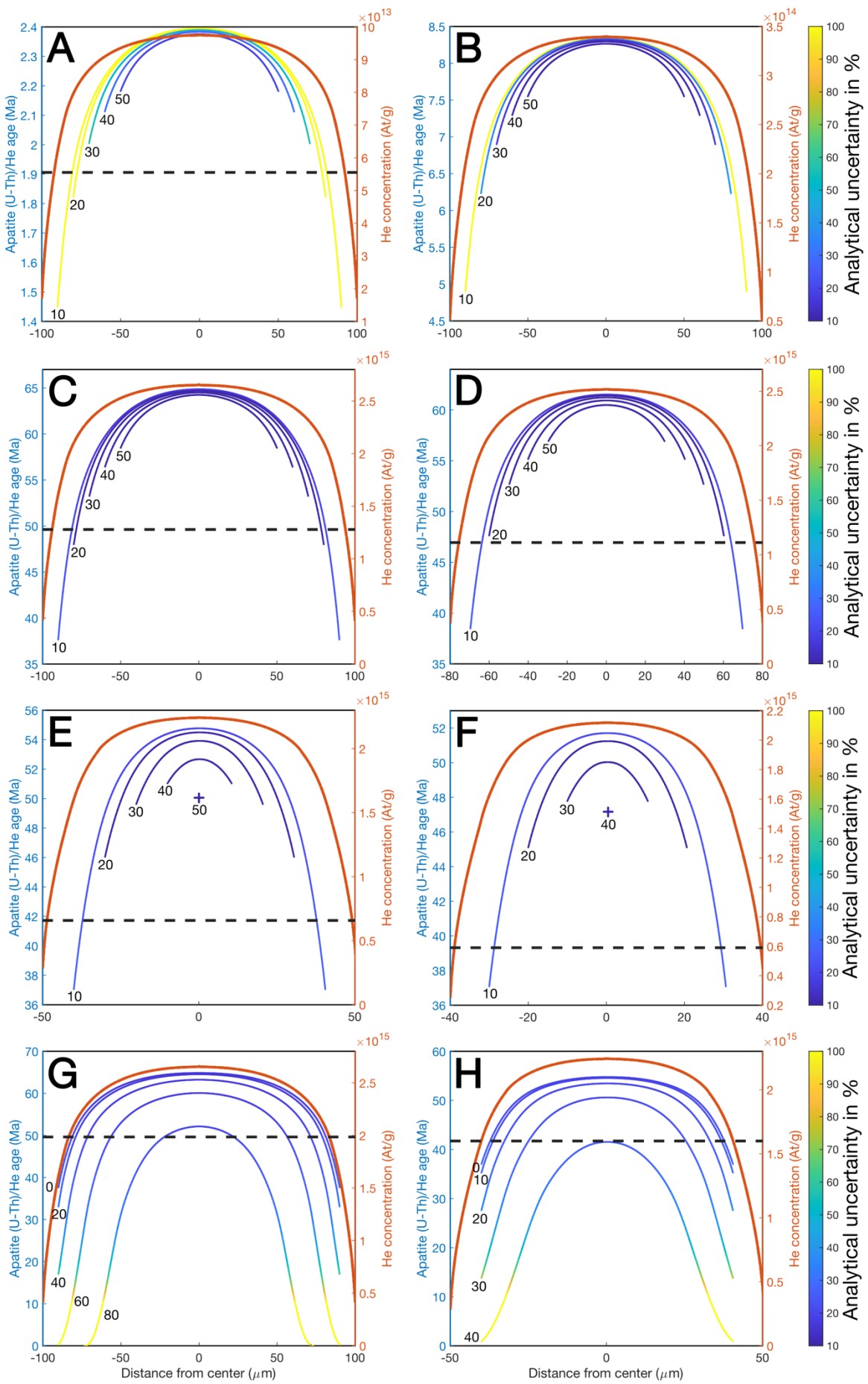

Fig. 8: Predicted in-situ apatite (U-Th-Sm)/He dates (blue to yellow lines) and He concentration profile (orange line) for an infinitely long, cylindrical-shaped apatite with homogenous radiogenic nuclide distribution (U, Th and Sm concentration of 10 ppm). Predicted dates are calculated by integrating the modelled He distribution over an entire ablation pit volume of variable diameters (black numbers on curves in A-H), which is

continuously measured across the grain. In reality, discrete (rather than continuous) pits would be measured, and smooth curves such as those shown here would not be possible. In-situ date profiles are colour-coded according to expected analytical uncertainties calculated with an observed standard deviation of the $^4$He blank of 0.000079 ncc. A) Model results assuming a constant cooling rate of 40°C/Myr to a final temperature of 10°C and a grain

radius of 100 μm. The corresponding whole-grain date for a sphere with a similar sphere-equivalent radius (radius*1.5) corrected for alpha ejection is 1.9 Ma. Modelled in-situ dates with variable spot diameters (10, 20, 30, 40 and 50 μm) range from 2.4 Ma in the centre of the grain to 1.45 Ma half the spot diameter away from the grain rim. B) Model results assuming constant cooling at 10°C/Myr to 10°C and a grain radius of 100 μm. The

corresponding whole-grain date corrected for alpha ejection is 6.5 Ma. Modelled in-situ dates with variable spot diameters (10-50 μm) range from 8.3 Ma in the centre of the grain to 4.8 Ma half the spot diameter away from the grain rim. C) Model results assuming constant cooling at 1°C/Myr to 10°C and a grain radius of 100 μm. The corresponding whole-grain date corrected for alpha ejection is 64.9 Ma in the centre of the grain to 37.7 Ma half the spot

diameter away from the grain rim. D,E,F) Same as C) but with a grain radius of 80, 50, and 40 μm. The smaller grain radius results in younger whole-grain dates (47, 42, and 39 Ma, respectively) and a stronger relationship between in-situ dates and distance of measurement towards the grain rim. G) In-situ dates for a grain radius of 100 μm and spot diameter of 10 μm. Dates have been calculated for the central plane, dividing the cylinder into two

symmetrical sides along the crystallographic c-axis (black number 0 - 0 μm in the r-direction of Fig. 2) and planes cutting the grain at 20, 40, 60, and 80 μm above/below the central plane. H) In-situ dates for a grain radius of 50 μm and spot diameter of 10 μm. Dates have been calculated for the central plane and at other r-planes 10, 20, 30, 40 μm.


### 3.2 Effects of radionuclide zoning

Without practical analytical measurement methods to quantify inner-grain variations in radionuclides, whole-grain analyses commonly use an apriori assumption of a uniform radionuclide distribution. The in-situ (U-Th-Sm)/He dating technique produces spatially resolved (albeit averaged over the ablation pit) measurements of U, Th, and Sm (e.g. Horne et al. 2016, Danišík et al., 2017). In-situ measurements can provide information about inner-grain radionuclide variations and potentially lead to a reduction in date variability when excluding grains with radionuclide variations or by taking into account heterogeneities in the radionuclides.

Ideally, however, the 2D-3D distribution of parent nuclide concentrations is mapped in grains, which is possible with a new generation of instruments, such as by mapping parent nuclide concentrations with laser-ablation inductively coupled plasma mass spectrometry (Chew et al. 2017), time-of-flight secondary ion mass spectrometry (North et al. 2022) or synchroton X-ray fluorescence tomography (Sousa et al. 2024). Measured 3D radionuclide patterns can be incorporated in available implementations of 3D modelling of He production, ejection and diffusion (e.g. Gautheron et al. 2012). Although this procedure would be ideal, it is computationally and analytically expensive, and, therefore not routinely applied. Efficient thermal history modelling of (U-Th-Sm)/He data requires 1D modelling of He production, ejection and diffusion, which can be combined with time-efficient single-spot LA-ICP-MS measurements. This approach should be used to identify, and exclude, grains with complex radionuclide variations.

Here, we have analyzed several hundred LA-ICP-MS measurements done in our lab at the University of Tuebingen, Germany. The depth-resolved radionuclide measurements in apatite and zircon demonstrate that radionuclide zoning is common (supplement data SD1). Zoning is more common and pronounced in zircons; ~30% of all analyzed zircons have a core-to-rim ratio <0.5 or >2, whereas this fraction is at ~10% for apatites (Fig. S4). Not accounting for radionuclide zoning results in erroneous Ft-correction factors and resulting whole-grain dates (e.g., Hourigan et al. 2005). Here, we use our updated production-ejection-diffusion model to calculate the relationship between whole-grain (U-Th-Sm)/He dates and radionuclide variations (Fig. 9).

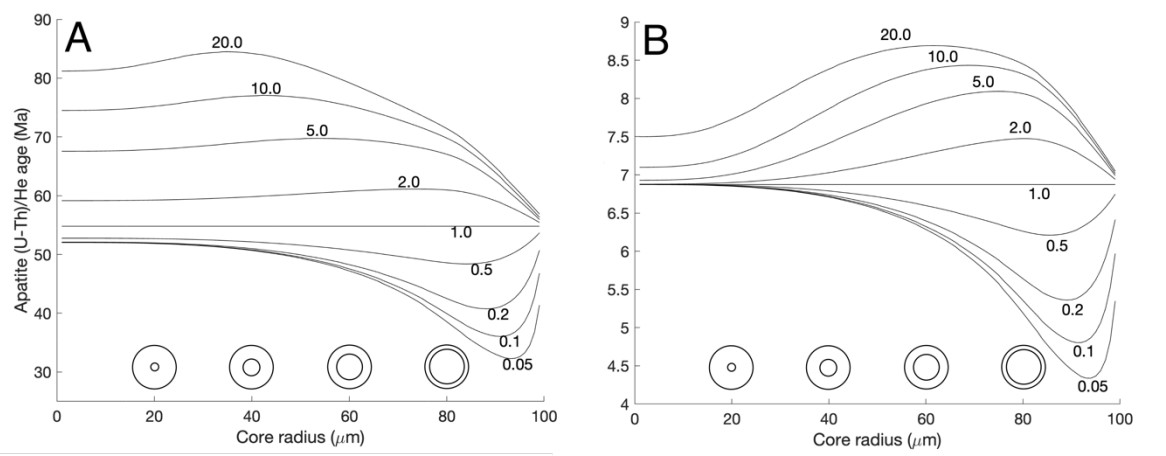

Fig. 9: Whole-grain apatite (U-Th-Sm)/He dates as a function of radionuclide variations (zoning). Isoline labels correspond to the core/rim ratio of radionuclides, assuming a single-step function in the concentration of U, Th, and Sm across the grain, where the x-axis specifies the location of the step in concentration. A) Dates for a cooling rate of 1°C/Myr and volume-averaged U, Th, and Sm concentration of 10 ppm. B) Dates for a cooling rate of 10°C/Myr and volume-averaged U, Th, and Sm concentration of 10 ppm.

Commonly observed core-to-rim ratios between 0.5 and 2 lead to ±10% date deviations (Fig. 9). Since observed radionuclide variations cannot be simplified with a single-step function (Fig. S3), we have scaled measured LA-ICP-MS derived depth variations to a grain radius of 100 μm and calculated whole-grain apatite and zircon (U-Th-Sm)/He dates for a cooling rate of 1 °C/Myr (Fig. 10). Single-grain dates are mainly a function of the mean eU of individual grains, but depending on the amount of radionuclide zoning, dates deviate from the corresponding date assuming homogenous radionuclide distribution (red line in Fig. 10). The correlation coefficient is 0.95 and 0.77 for all apatites and zircons, respectively, or, in other words, 5% and 23% of the variability in dates is the result of radionuclide zoning. Individual samples usually involve fewer grains with variations in dates caused by radionuclide zoning ranging from 1 to 40% and 19 to 84% for analyzed apatites and zircons (Fig. 10), respectively. In samples with a low variation in eU and strong radionuclide zoning, the majority of variability is caused by zoning and there is no significant relation with eU. For example, in the analyzed Fish Canyon tuff zircons, 84% of the variations in dates is due to zoning (Fig. 10B). As mentioned earlier, additional age dispersion can come from crystal fragmentation, radionuclide-rich inclusions, fluid inclusions and implantation of He from the exterior (e.g. Brown et al., 2013; Vermeesch et al., 2007; Spiegel et al., 2009; Danišík et al., 2017). It is, therefore, more likely to have overdispersed whole-grain (U-Th-Sm)/He dates,

and perfect relations between dates and eU will be the exception rather than the rule (Flowers
et al., 2022).

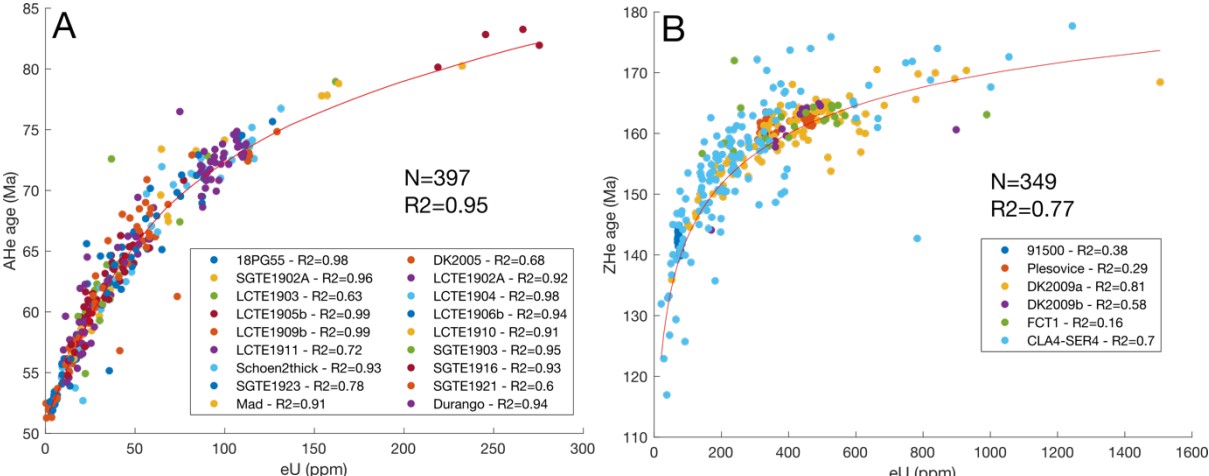

Fig. 10: Simulated apatite (A) and zircon (B) (U-Th-Sm)/He dates for a constant cooling rate
of 1°C/Myr, and measured grain-specific U, Th, and Sm variations (coloured dots).
Radionuclide variations were measured in age standards and random samples with a LA-ICP-
MS system and scaled to a common grain size of 100 μm, assuming symmetric zoning
around the c-axis. This was used to model grain-specific whole-grain (U-Th-Sm)/He dates.
The red line represents the relation between dates and homogenous radionuclide distribution.
The correlation coefficient for the whole dataset and individual samples are shown. Note that
the $1-R^2$ is the fraction of spread caused by radionuclide zoning.


The observed radionuclide variations and resulting date dispersion in Figure 10 allow for
estimating the minimum sample size required to reach a defined correlation coefficient
between date and eU. We did this by randomly sampling 20,000 times 3, 4, 5 …30 grains
from our database and determined the relationship between the correlation coefficient and
sample size and found that a minimum of 10 apatite grains is needed to reach an $R^2$ of 0.8,
and an impractical 23 whole-grain ZHe dates are theoretically needed to reach a minimum $R^2$
of 0.5 (see supplement data SD3 for details).
In summary, whole-grain (U-Th-Sm)/He age variations with eU are often biased by
radionuclide zonation. In in-situ (U-Th-Sm)/He studies, radionuclides are typically
determined from a single pit, several tens of micrometer deep, drilled and analyzed using LA-
ICP-MS (e.g., Pickering et al., 2020). After applying a downhole fractionation correction
(Paton et al., 2010), depth-resolved radionuclide profiles in apatite and zircon grains enable
the identification of zoned grains, which should be excluded from further analyses. Since this
analytical step is mandatory for the in-situ (U-Th-Sm)/He method, single-grain data should,

in theory, lead to less dispersion in date vs. eU plots and likely also produce more reliable

thermal history reconstructions. In contrast, including grains with identified radionuclide

zoning is likely to produce inaccurate results, as helium is generated and measured from

different volumes within the grain (e.g., Vermeesch et al., 2023).

3.3 Thermal history modelling of in-situ (U-Th-Sm)/He data

The relative distribution of He within an apatite or zircon grain is a function of the

distribution of radionuclides, grain morphology, and the cooling history. A suite of whole-

grain analyses can be used to reconstruct potential cooling histories under the precondition

that analyzed grains have different grain sizes and/or eU (e.g., Ketcham, 2005; Flowers et al.,

2009; Gautheron et al., 2009; Guenthner et al., 2013). This approach, however, has the risk of

including grains with internal variations in radionuclides, and is, therefore, often applied in

combination with other thermochronometric systems (e.g., apatite fission track data). Similar

to whole-grain analyses, radionuclide zonation can bias the interpretation of cooling histories

derived from the apatite $^4$He/$^3$He method (Farley et al., 2010), which indirectly measures the

He profile by a stepwise degassing of He from proton-irradiated apatite grains.

Thermal history modelling with in-situ (U-Th-Sm)/He data could be done by (i) measuring

multiple grains that vary in size and/or eU similar to the whole-grain approach, or (ii)

reconstructing the He profile with multiple measurements in a single grain comparable to the

$^4$He/$^3$He method (e.g. Danišík et al., 2017). Both approaches are applied in the following to

reconstruct common cooling paths from synthetic datasets. A robust methodology requires

knowing or estimating (i) the grain geometry, (ii) the position of the in-situ measurements

within the grain, (iii) the radionuclide distribution within the grain, and (IV) building an

appropriate model to account for the previous factors.

In theory, complex grain morphologies could be used for such an approach, but this would

require implementing grain-specific 3D models. Thermal history modelling with 3D models

is time-consuming and, therefore, not practical for routine analysis. Similar to whole-grain

analyses, it is therefore recommended to make in-situ measurements of grains with simple

geometries characterized by straight and/or 2D-3D constant curvatures such as spherical,

elliptical, and cylindrical shapes. Preferably, the in-situ measurement can be approximated

with a 1D modelling approach similar to whole-grain and $^4$He/$^3$He analyses, where the

sphere-equivalent radius has been shown to be a good approximation (e.g. Meesters and

Dunai, 2002; Farley et al., 2010). Unlike whole-grain and [4]He/[3]He analyses, the in-situ method requires modelling the He concentration within the ablated pit volume.

We conducted three different measurement approaches to evaluate the utility of in-situ dating techniques for thermal history reconstruction. First, a set of two cylindrical spots with 30 μm diameter and 5 μm depth in the centre of cylindrical-shaped grains with radii of 80 and 40 μm and similar eU were forward-modelled with three different cooling histories: (1) a constant cooling rate of 1°C/Myr, (2) rapid cooling of 100°C/Myr at 60 Ma to surface temperature
followed by no cooling, and (3) a step-increase in cooling rate from an initial 1°C/Myr until 10 Ma to 50°C, followed by 4°C/Myr cooling to surface temperature. Analytical uncertainties of theoretical He measurements were calculated with Eq. 21 and lab-specific blank level $(2.11 \times 10^6$ atoms), resulting in uncertainties of ~8 % for cooling histories (1) and (2) and ~15% for cooling history (3). Several thousand forward models were conducted, and the
goodness-of-fit (GOF) of predicted cooling paths was determined. We used the same definition of the GOF and colour scheme as used in HeFTy (Ketcham, 2005). Good and acceptable model paths retrieve the input He profile and cooling paths especially in the center of grains, while modelled He concentrations deviate in the outer 20 μm for some/most cooling paths (Fig. 11).

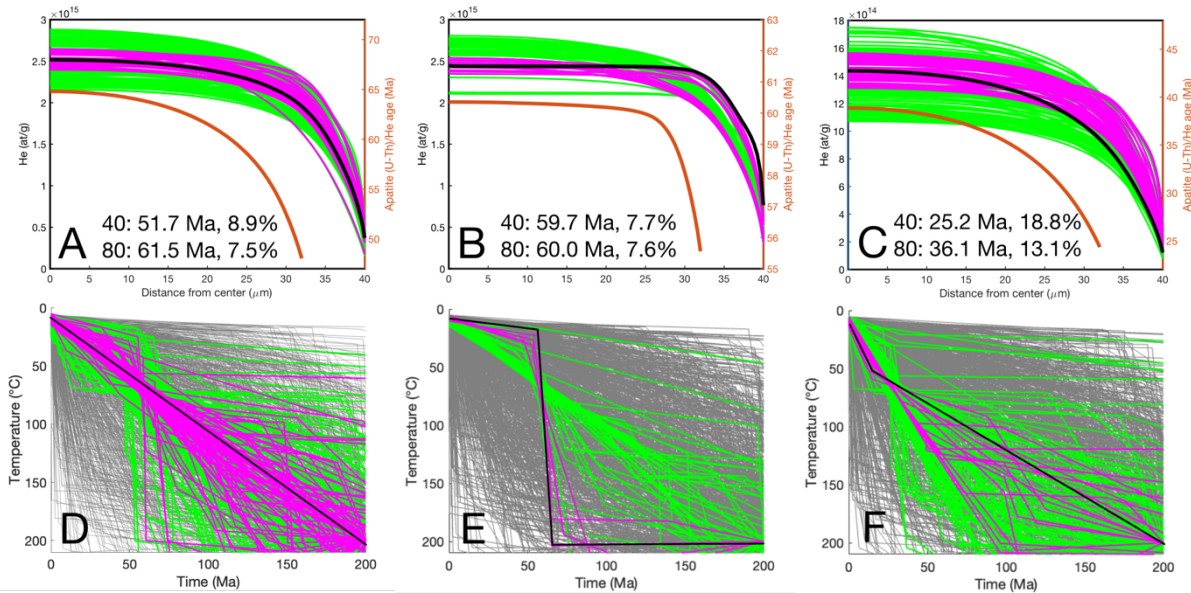


Fig. 11: Modelling of cooling histories for three synthetic datasets with two laser-ablation spot measurements in apatite grains with a pit diameter of 30 μm (5 μm depth), a grain radius of 80 and 40 μm and U, Th, and Sm concentrations of 10 ppm. Upper panels (A,B,C) show the synthetic (black line) and modelled (green, magenta) He profiles, while the brown line
represents the in-situ (U-Th-Sm)/He dates for the 40 μm grain. The lower panels (D,E,F)

show the input (black) and modelled (gray, green, magenta) cooling paths. Predicted cooling histories with acceptable paths are green (GOF>0.05), good paths are magenta (GOF>0.5), and paths with a GOF<0.05 are grey. A,D) Data was calculated with a constant cooling rate of 1°C/Myr. B) Input data were modelled with rapid cooling at 60 Ma to surface temperature. C) Initial slow cooling with 1°C/Myr to 50°C at 10 Ma is followed by faster cooling to the surface with 4°C/Myr.

Second, a set of synthetically generated He measurements were taken along a profile in a single grain. We use a cylindrical grain with a radius of 100 μm and the same thermal histories as in the previous experiment. We sampled the He profile with an assumed cylindrical spot with a diameter of 20 μm and depth of 5 μm at five locations from the centre of the grain to the rim. The resulting He profiles and synthetic He measurements with uncertainties are shown in Fig. 12A-C. Analytical uncertainties of theoretical He measurements were calculated with Eq. 21 and lab-specific blank level ($2.11 \times 10^6$ atoms), resulting in uncertainties of 16-20 % for cooling history (1) and (2) and ~26-40% for cooling history (3). Most acceptable cooling histories overlap or are close to the input parameters, suggesting that in-situ (U-Th-Sm)/He measurements within a single grain can be used to get information on its cooling history (Fig. 12D-F).

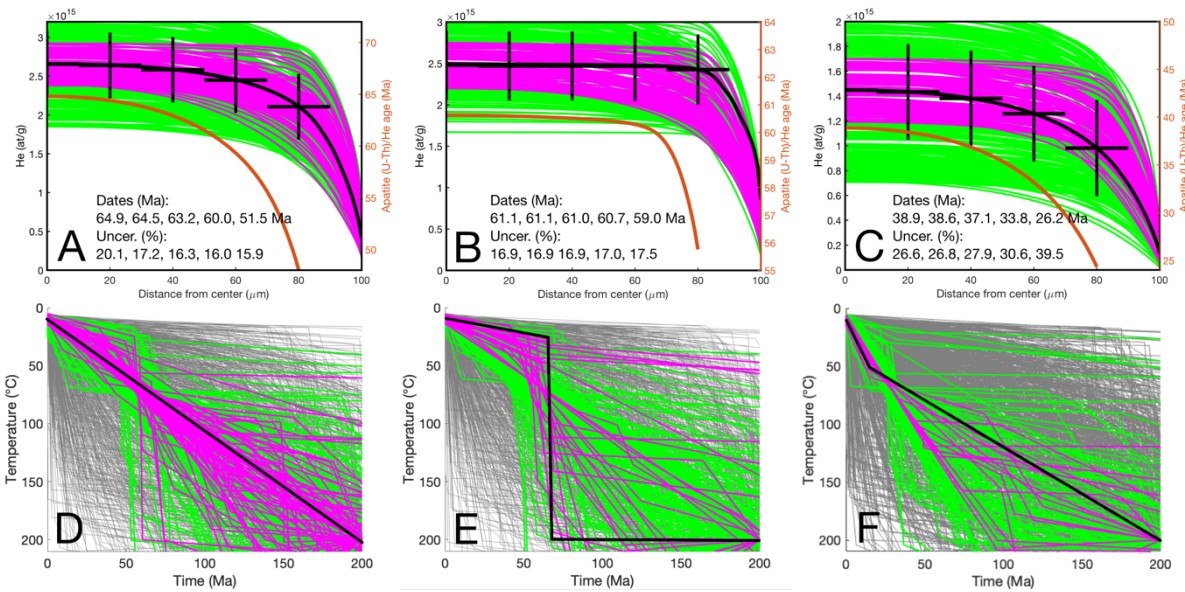

Fig. 12: Cooling histories predicted from in-situ (U-Th-Sm)/He measurements sampled along the He profile of a synthetic cylindrical apatite grain with a radius of 100 μm, and U, Th, and Sm concentrations of 10 ppm. Five 20 μm diameter (5 μm depth) ablation pits across the grain (horizontal black lines) are used as synthetic input data. Upper panels (A,B,C) do show synthetic (black line) and modelled (green, magenta) He profiles, while the brown line

represents the in-situ (U-Th-Sm)/He dates. Resulting in-situ (U-Th-Sm)/He dates and He uncertainties are also given from the center (left) to the rim (right). The lower panels (D,E,F) show the input (black) and modelled (gray, green, magenta) cooling paths. Predicted cooling histories with acceptable paths are green (GOF>0.05), good paths are magenta (GOF>0.5), and paths with a GOF<0.05 are grey. Data modelled with a constant 1°C/Myr cooling rate

(A,D), a rapid cooling event at 60 Ma to surface temperature (B,E), and slow cooling with 1°C/Myr to 50°C followed by faster cooling to the surface with 4°C/Myr from 10 Ma (C,F).

Although we would not recommend interpreting grains with internal radionuclide variations, here we investigate the consequences for in-situ and whole-grain thermal history modelling.

We assume a scenario in which the outer 10 μm of grains is enriched in radionuclides (U, Th, Sm: 50 ppm) compared to the grain interior (U, Th, Sm: 10 ppm). This is a nasty scenario, resulting in largely underestimated whole-grain (U-Th-Sm)/He dates if not corrected for (Fig. 9). Analogous to the previous thermal history modelling, a fast cooling from high temperature to surface temperature at 60 Ma was used to produce theoretical (U-Th-Sm)/He

data for cylindrical grains with 100, 80 and 40 μm radii.
The general cooling trend can be retrieved in case the radionuclide distribution is precisely known and the He profile is sampled with several measurements in a single grain (Fig. 13A,B) or multiple grains with variable sizes are analysed (Fig. 13G-J). We also tested the inversion performance assuming a homogenous radionuclide concentration of 10 ppm,

measured for instance with a LA-ICP-MS pit in the centre of the grain (not reaching the grain rim). In addition, a homogenous radionuclide concentration of 17.6, 19.4 and 27.5 ppm for grains with 40, 80 and100 μm radii was used as input, representing the grain averaged concentration, as measured through whole-grain analyses. As expected, observed inner-grain He variations, with increased concentrations toward the grain rim, are impossible to model

with a constant radionuclide concentration (Fig. 13C-K). The He concentrations in the centre of the modelled grains with radii >40 μm, is nearly unaffected by the high radionuclide concentration in the rim of the grain (e.g., Fig. 13A,G). In this specific scenario, modelling in-situ (U-Th-Sm)/He data from the centre of grains correctly retrieves the cooling assuming constant radionuclide concentration of 10 ppm (Fig. 13I,J). Instead, using the mean whole-

grain radionuclide concentration as input results in incorrect cooling histories (Fig. 13K,L). We modelled whole-grain (U-Th-Sm)/He data to investigate the ability to reconstruct the input thermal history of zoned grains. Modelled whole grain (U-Th-Sm)/He are between 57.3 Ma, 58.7 Ma and 59.0 Ma for grains with 40, 80 and 100 μm grain radius and rapid cooling

to the surface temperature at 60 Ma. The whole-grain data can retrieve the general cooling trend in case the radionuclide distribution is precisely known (Fig. 13M). The latter, however, is commonly not measured, and instead, the whole-grain average radionuclide concentration is measured and used for thermal history modelling. Interestingly, the modelling does not retrieve the correct or any acceptable cooling history (Fig. 13O), which we interpret as a result of incorrect He diffusivities associated with the assumption of homogenous inter-grain variable radionuclide concentrations (19.4 vs. 27.5 ppm). For comparison, we also modelled the thermal history using a radionuclide concentration of 10 ppm for both grains (Fig. 13N). Although acceptable and good thermal paths are predicted by inverse modelling, the correct input thermal history could not be retrieved.

# Single-grain approach

### Core/Rim - 10/50 ppm    Constant - 10 ppm    Constant - 17.6 ppm

# Multi-grain approach

### Core/Rim - 10/50 ppm    Constant - 10 ppm    Constant - 19.4/27.5 ppm

# Whole-grain approach

### Core/Rim - 10/50 ppm    Constant - 10 ppm    Constant - 19.4/27.5 ppm

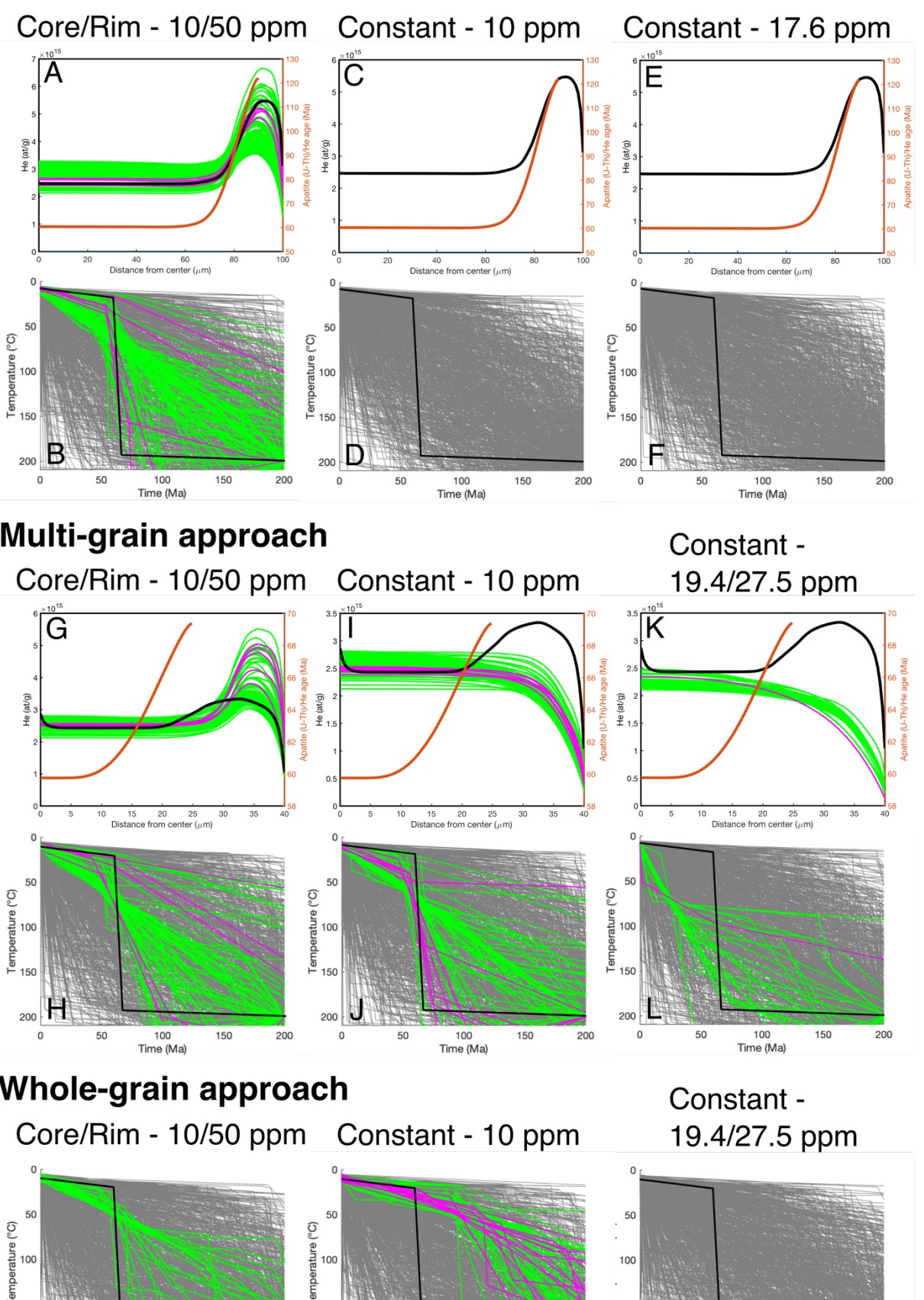

Fig. 13: Cooling histories predicted from in-situ and whole-grain (U-Th-Sm)/He measurements of a synthetic cylindrical apatite grain with a radius of 40, 80 and 100 μm, and U, Th, and Sm concentrations of 10 ppm in the core and 50 ppm in the 10 μm wide rim. All input data is modelled with fast cooling to surface temperature at 60 Ma. Retrieving the thermal history assumes either (i) precise knowledge of the radionuclide variation and distribution (left column), (ii) a homogenous radionuclide concentration of 10 ppm (middle column) and (iii) whole-grain average homogenous radionuclide concentration (right column). A-F) Single-grain approach with five 20 μm diameter (5 μm depth) ablation pits across a grain with 100 μm radius are used as synthetic input data. Upper panels (A,C,E) do show synthetic (black line) and modelled (green, magenta) He profiles, while the brown line represents the in-situ (U-Th-Sm)/He dates. The lower panels (B,D,F) show the input (black) and modelled (gray, green, magenta) cooling paths. Predicted cooling histories with acceptable paths are green (GOF>0.05), good paths are magenta (GOF>0.5), and paths with a GOF<0.05 are grey. G-L) Multi-grain approach using a central 30 μm diameter (5 μm depth) ablation pit in two grains (40 and 80 μm radii) as synthetic input data. G-L) Similar to A-F but for a multi-grain approach. M-O) Thermal inversion results for two grains with 40 and 80 μm radii using the whole-grain approach.

## 4.0 Discussion

4.1 Synthesis of results

The previous results suggest that in-situ (U-Th-Sm)/He dating can provide an improvement in date and thermal history calculation compared to the conventional whole-grain analyses. This is due to the technique's capability to detect for radionuclide zoning, thereby resulting in reliable date predictions and thermal history reconstructions. The latter, however, can be only achieved when grains with radionuclide zoning are excluded, since accounting for zoning would ideally require a 3D mapping and modelling approach which is not routinely feasible. However, a caveat of the in-situ approach is that individual spot dates will be variable across the grain even without radionuclide zoning, and a framework is required for interpreting them.

Based on the previous analysis, we suggest two different measurement approaches for in-situ (U-Th-Sm)/He dates to yield geologically relevant data. These approaches include single-spot

measurements from multiple grains from a single sample and multiple spot locations across a

single grain. In both cases, potential inner-grain radionuclide variations need to be studied,

for instance, making maps, line scans or drilling through the whole grain with a LA-ICP-MS

system. In addition, a combination of both single- and multiple-spot approaches might be

practical, with single-spot measurements in small grains and multiple spots in larger grains.

Anyway, the resulting dates can be used to reconstruct the sample's cooling history for

cooling rates between 1-40 °C/Myr. Faster cooling rates (e.g., 100 °C/Myr) characteristic of

rapidly exhuming orogens (e.g., Himalaya, Taiwan, New Zealand) were not explored in this

study and may present additional challenges if parent radionuclide concentrations are low

(e.g., 1-10 ppm) lending to low He concentrations that are below the detection limit using

reasonable pit diameters (<<100 μm).

Results presented here were based on simulated ablation pit diameters of 20 and 30 μm (5 μm

deep) and U, Th, and Sm concentrations of 10 ppm. With these values, in-situ dating of

apatite grains as young as ~60 Ma and analytical uncertainty of ~10% is possible (with our

measurement limit of detection being 0.000079 ncc He). Accordingly, ages as young as 10

Ma can be measured with a pit diameter of 30 μm (10 μm deep). Increasing the pit volume

further would be problematic for deriving the cooling history from in-situ (U-Th-Sm)/He

data, especially if grains are small. Larger pit volumes integrate more likely areas of the grain

affected by He ejection and limit the number of pits placed in a single grain. Given these

factors, we recommend that future investigations of in-situ analytical procedures analyse

large grains and measure He in as small as detectable pit volumes for reconstructing thermal

histories.

4.2 Meaning of in-situ dates

Whole-grain (U-Th-Sm)/He dates primarily depend on the sample cooling history and, to a

lesser degree vary with grain size, radionuclide concentration and the alpha-damage density.

In addition, they can occasionally be biased by radionuclide zoning or inclusions (e.g.,

Farley, 2002). In the rare case of rapid cooling to surface temperatures, the whole-grain date

(irrespective of grain size and radionuclide concentration) reflects the time of that cooling

event (e.g., Wolf et al., 1998). Importantly, the same date can be reproduced by slow

monotonic cooling and even cooling followed by reheating (e.g. Wolf et al., 1998). In the

latter case, the (U-Th-Sm)/He date might even correspond to the time when the sample was at

the surface temperature. A single whole-grain (U-Th-Sm)/He date alone does not hold much

information on the thermal history, which requires analysis of more grains or a joint interpretation with other thermochronology methods and geological constraints.

Our modelling in-situ and whole-grain (U-Th-Sm)/He dates for slow to fast cooling rates (1-40°C/Myr) indicates that dates are commonly older than the corresponding whole-grain date for monotonic cooling (Fig. 8). In a study with a larger variation in parameters, we explored the relationship between whole-grain and in-situ dates for very slow to fast cooling rates (0.5-40°C/Myr) in more detail (Fig. 14). Monotonic cooling, irrespective of the cooling rate,

results in roughly 30% older in-situ dates compared to whole-grain dates (Fig. 14A,B). Cooling with a rate of 10°C/Myr to a surface temperature of 10°C at different times results in variable differences in whole-grain vs. in-situ dates (Fig. 14C-D). Dates are nearly identical for cooling to surface temperature at 50 Ma, and dates diverge for cooling histories to surface temperatures at younger times.

The fundamental difference between whole-grain and in-situ (U-Th-Sm)/He dating is the location and volume where He is measured in a grain, whereas the differences in dates between the methods strongly depends on the cooling history and associated diffusion history. He production and ejection result in strong concentration differences in grains, which set the pace for diffusional He loss increasing from the centre to the rim of a grain, as

illustrated with our modelled He profiles (e.g. Fig. 12). Measuring He in the centre of grains, as is common practice in in-situ dating, leads to older ages than whole-grain dating. The latter includes diffusion-related He-depleted grain rims, yielding younger dates. Samples where the majority of produced He has not been affected by high diffusion rates have similar whole-grain and in-situ dates, such as in the rapidly cooled Fish Canyon age standard (e.g. Horne et

al., 2016; Pickering et al., 2020) or Ellendale pipe samples (Evans et al., 2015). In one additional scenario, whole-grain and in-situ dates are anticipated to exhibit identical dates. This occurs with very-large crystals irrespective of their specific cooling history, exemplified by Durango apatite and Madagascar monazite and zircon (see Boyce et al., 2006; Evans et al., 2015; Horne et al., 2019; Vermeesch et al., 2012).

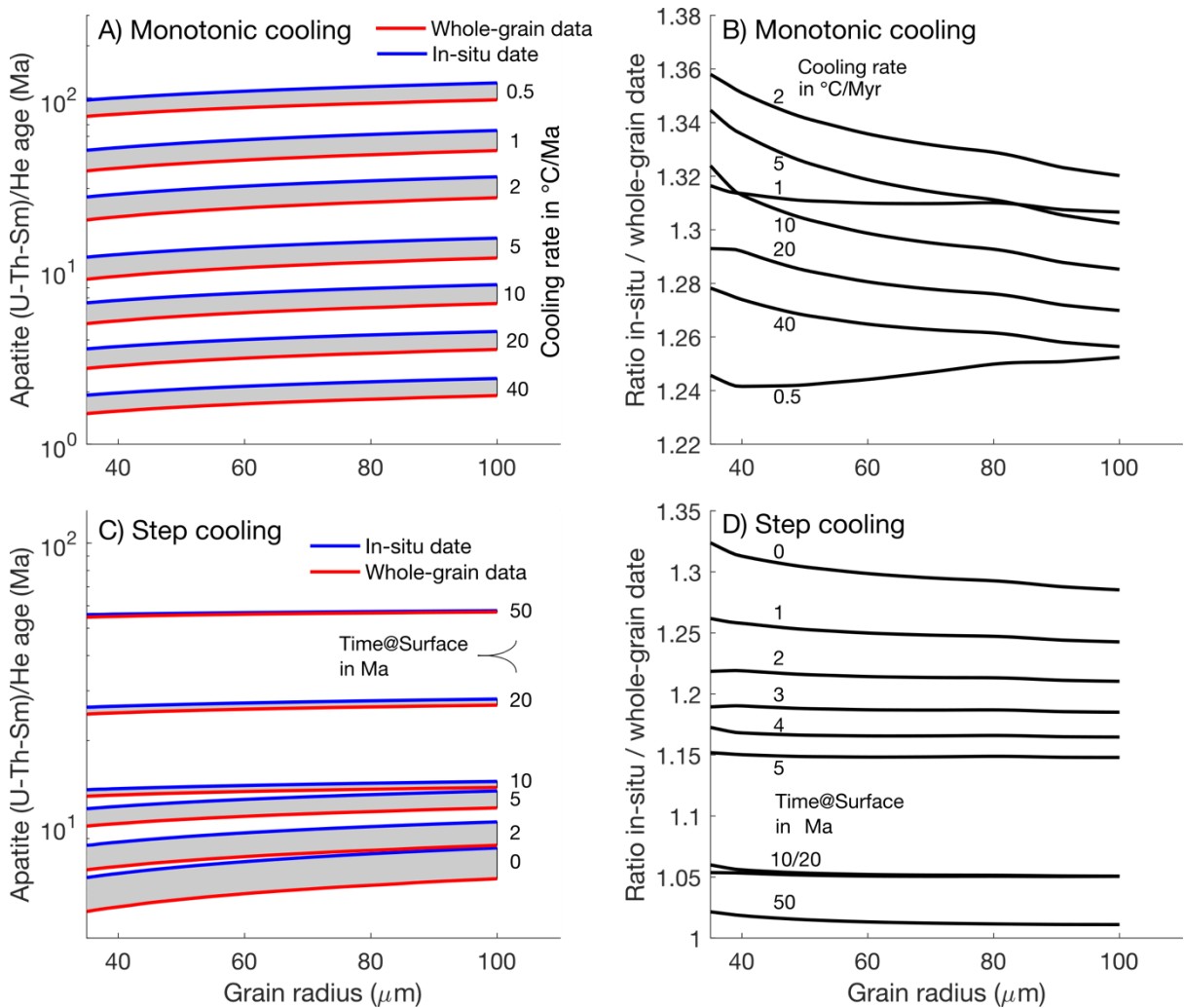

Fig. 14: Whole-grain vs. in-situ (U-Th-Sm)/He dates as a function of grain size and cooling rate for a U, Th, and Sm concentration of 10 ppm (homogenously distributed) and assuming a single spot radius of 10 μm (2 μm deep) in the centre of a spherical grain. A) Dates for cooling rates of 0.5 to 40 °C/Myr. B) Ratio of in-situ and whole-grain dates as a function of grain size and cooling rate. C-D) Same as A and B, but modelled with a step cooling to surface temperature (10°C) at different times (0-50 Ma) with a cooling rate of 10°C/Myr.

Tripathy-Lang et al. (2013) applying the in-situ method to detrital zircons from a tributary of the Indus River in the Himalayas, draining the southern part of the Ladakh batholith, with the in-situ method. Cooling of the Ladakh batholith through the He partial retention zone for zircons likely occurred rapidly in Oligoence times (Kirstein et al., 2009). According to our modelling results, this should result in similar whole-grain and in-situ zircon dates. In fact, the resulting whole-grain and in-situ date distributions show comparable patterns, with slightly older whole-grain dates (mean date 29 vs. 26 Ma). Tripathy-Lang et al. (2013)

interpreted the difference to result from preferential grain selection for whole-grain analyses and considered the in-situ dates to be more representative. Alternatively, the larger spread and slight shift to older dates may be based on methodological differences, where in-situ dates are generally older for samples that have experienced diffusional He loss.

In summary, in-situ (U-Th-Sm)/He dating of apatite/zircon is an alternative to whole-grain dating with obvious advantages. However, our modelling results demonstrate that dates cannot simply be interpreted together with whole-grain data. If analysed grains are expected to have lost a significant fraction of He by diffusion, in-situ dates will be older than whole-grain dates. In the case of bedrock studies, in-situ data can be interpreted using modified

thermal models introduced here, and to aid comparisons to existing whole-grain datasets, corresponding whole-grain data can be derived from those models. In most detrital studies, it is impossible to know the fractional loss of He of each individual grain and in-situ dates, if measured in the centre of grains, will be systematically older. In case the dates of source areas are largely different (e.g., 15 vs. 30 vs. 90 Ma), inferences from in-situ (U-Th-Sm)/He

dating might still be acceptable, such as the detrital zircon study from the Inn River in the European Alps by Dunkl et al. (2024).

4.3 Grain selection considerations for in-situ measurement


Grain selection for the in-situ (U-Th-Sm)/He method follows criteria similar to the whole-grain method. Simple 1D thermal history modelling requires that selected grains have smooth surfaces and are symmetrical, such as spheres and cylinders. The long-prismatic shape and basal cleavage direction often result in the fragmentation of apatite grains, especially during

the mineral separation process (e.g., Farley, 2002). Interpreting apatite fragments with the whole-grain and $^4$He/$^3$He method usually requires corrections for grain fragmentation (e.g., Brown et al., 2013; Flowers and Farley, 2012). Instead, fragments of apatite grains broken along the basal faces can be treated similarly to intact grains with the in-situ (U-Th-Sm)/He method.

Grains should be free of inclusions to avoid excess He from long-alpha stopping distances (e.g., Farley, 2002). The pre-measurement exposure of the inner surface facilitates thorough inspection of the grain interior and identification of potential inclusions at sub-μm resolution using 1000x magnification. Even though not visually evident with microscopy, inclusions can be identified by measuring radionuclide concentrations with LA-ICP-MS. The downside of

analyzing inner surfaces after abrasion is that roughly half of the grain is not available for inspection, and thus outliers related to excess He from mineral inclusions (abraded away) will still be an issue in in-situ (U-Th-Sm)/He dating.

Similar to the whole-grain method, reliable date determination and thermal history reconstructions require precise measurements of grain geometries (e.g., Glotzbach et al.,

2019). Future applications of the in-situ (U-Th-Sm)/He methodology will show if geometry measurements are required before embedding grains, or measurements can be done within the mount following grain selection. Measurements of the distance between He laser pits and grain prism faces can follow pit volume measurements. An issue that may arise is the complexity involved in accurately determining the position of the inner surface relative to the

original grain boundary, particularly in the vertical dimension of mounted grains. The common tetragonal and hexagonal cross-sectional shapes of zircons and apatites result in theoretically variable-sized inner surfaces (e.g. Fig. 7). A symmetrical apatite and simple zircon grain have a ratio between circum- to inner radius of 1/1.15 and 1/1.41, respectively. It is, therefore, mandatory to accurately determine the correct location of the pit location with

respect to the whole-grain geometry.

4.4 Recommended reporting procedure for in-situ analytical data.

We recommend using the 1D modelling approach only for grains with homogeneous or

concentric radionuclide distribution. The latter should be verified by spatial-/depth-resolved radionuclide information, e.g., with LA-ICP-MS depth profiling or mapping. Based on the model results presented here and the discussion in the previous section, we recommend reporting several different aspects of in-situ measurements. These items will enable not only reproduction of dates for each spot, but also facilitate modelling of grain thermal histories

using the software of this study. Essential items to report in data tables for each grain include: 1) grain geometry (preferably with photos in a supplement) and assumed grain geometry (e.g., sphere, infinite cylinder, other) used for age calculation, 2) (for each ablation pit across a grain) the pit diameter, measured volume, depth, and center point of the pit relative to the a-, b- and c-axis of the grain, 3) the He measured from the ablation pit, 4) the U, Th and Sm

concentration profiles, 5) the calculated in-situ grain date, and 6) the whole-grain equivalent date (which requires thermal history modelling, see Fig. 14). Reporting of the above information enables thermal history modelling of individual grains and comparison of in-situ dates to whole-grain dates from neighbouring areas and/or previous studies.

4.5 Future considerations

Although the theoretical benefits and limitations have been explored here, more applications
of the in-situ (U-Th-Sm)/He method to samples are required. Future studies should explore (i)
the spatial relationship between radionuclide zoning and resulting He distribution, and (ii) the
reliability of in-situ (U-Th-Sm)/He-derived thermal history reconstructions. Lastly (iii), as
previously mentioned, future modelling studies should evaluate tradeoffs between the cooling
rate (particularly at higher cooling rates of >10 °C/Myr) and parent radionuclide
concentrations to evaluate the limits of in-situ dating to produce geologically interpretable
data.


**5.0 Conclusions**

This study examined the theoretical relationship between the parent radionuclide distribution
and the resulting He concentrations within a grain (such as apatite or zircon). This was done
using an updated version of the production, ejection, and diffusion model (i.e., RDAAM). We
investigated the dependencies of predicted whole-grain and in-situ apatite and zircon (U-Th-
Sm)/He dates for monotonic cooling histories (1-40 °C/Myr), grain size (40-100 μm), and (in
the case of in-situ data) the position of the measurement within the grain. In addition, we
explored strategies for reconstructing the thermal history from multiple and single apatite
grains.
Model predictions revealed that the He concentration and resulting in-situ date is mainly a
function of the grain size, eU, and distance to the grain rim. Thus, the interpretation of in-situ
(U-Th-Sm)/He dates necessitates the assessment of the grain geometry of the measured grains
and determining the distance between the laser spot and the closest prismatic face. Most
importantly, in-situ dates for samples that experienced diffusional He loss will be older than
whole-grain dates. In most cases, understanding in-situ data necessitates the application of
adapted thermal models such as those introduced in this study. Additionally, to facilitate a
comparison to existing whole-grain data, corresponding whole-grain dates can be determined
through thermal history modelling.
Our observations revealed that radionuclide zoning is not an anomaly but a prevalent
occurrence in both apatite and zircon. Analysis of a substantial dataset using LA-ICP-MS for

radionuclide measurements in these minerals demonstrated that the observed radionuclide zoning has, if disregarded, the potential to substantially skew the relationships between effective uranium (eU) and whole-grain dates. Furthermore, results suggest that a minimum of 10 apatite grains are needed to reach an $R^2$ of 0.8 between eU and date and a labour-intensive number (23) of whole-grain ZHe dates is needed to reach a minimum $R^2$ of 0.5 between eU and date.

Two promising approaches exist for reconstructing the thermal history of rocks using the in-situ (U-Th-Sm)/He method. Similar to data obtained from whole grains, variations in grain size and/or effective uranium content, which lead to differences in helium diffusivity and in-situ dates, can be utilized for thermal history reconstructions. The in-situ (U-Th-Sm)/He method can measure a He concentration profile in single grains, which is, among other factors, controlled by the cooling history. Modelling results suggest that several in-situ (U-Th-Sm)/He measurements along a profile from the centre of a grain to the prism face can be inverted to reconstruct the thermal history of a single grain.

**Author contribution**

Conceptualization: CG, TE

Formal analysis: CG

Investigation: CG

Methodology: CG

Software: CG

Visualization: CG

Writing: CG, TE

**Competing interests**

The contact author has declared that none of the authors has any competing interests.

**Acknowledgements**

This study was supported by (1) the Deutsche Forschungsgemeinschaft (DFG) to Christoph Glotzbach (GL 724/11-1) under the priority program 4D-MB and is a contribution to the AlpArray initiative, (2) the Bundesgesellschaft für Endlagerung (BGE – STAFuE-21-12-Klei), and (3) a large equipment funding by the DFG to Todd Ehlers (INST 37/1041-1 FUGG). The manuscript benefitted from discussions with Sarah Falkowski and Ann-Kathrin

Maier. The modified version of RDAAM and ZRDAAM software used in this study is available from the Zenodo repository (**https://zenodo.org/records/10531763**).

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
