# Peer review of "Interpreting cooling dates and histories from laser ablation in-situ (U-Th-Sm)/He thermochronometry: A modelling perspective."

_Geochronology, 2024_

## Referee Comment (RC2)

[referee-annotated manuscript omitted]

---

## Author Response (AR1)

**Reply to reviewers comments – Geochronology (GCHRON-2024-12)**

We would like to thank both reviewers and the associate editor for their comments and corrections on the original manuscript. Most of the more significant comments deal with updating figures and better explaining our modeling approach. Find below our point-by-point replies to all comments.

Abbreviations used:

RC1: Comments from reviewer 1

RC2: Comments from reviewer 2

AEC: Comments from the associate editor

AR: Author response

**RC1:** The authors mention that approaching of the zonation by a two shell model is a simplification. That was for me the most disturbing issue while reading the manuscript. In the every-day practice we see irregular, chaotic patterns and less frequently concentric zonations. The CL patterns and the Raman or LA mapping confront us with the simplified two shell model.

**AR:** We do agree that this two- or multi-shell model is a simplification required to do thermal history modeling in a reasonable time. Full spatial-variable 3D modeling is not practical and would also require additional information than 2D mapping approaches, which also do not completely record the full 3D information. There were approaches presented during the last thermochron conference in Italy. We added references to more sophisticated 3D analytical and modelling strategies in the method and result section. In addition we added a discussion on the limitations in thermal history modeling that we are facing in the thermochronological community.

**RC1:** Line 33: "In most cases, resulting in-situ dates are approximately 30% older than corresponding whole-grain dates, except for samples exhibiting negligible diffusional helium loss." This statement is in this form misleading. Such a general percentage can't be determined, it needs explanation, in which conditions calculated the authors this value. Or when the length of the Abstract does not allow, then better to delete this sentence that can easily be the starting point of a misused, and endlessly recycled "urban legend".

**AR:** We do think that this is a fundamental difference between whole grain and in-situ dating and should be stated in the abstract, but to prevent misuse we added details on the conditions under which this will occur.

**RC1:**

Line 444: alph --> alpha – **AR:** corrected

Line 492: "other r-planes" Explain it, please. – **AR:** For easier understanding, we replaced r-planes with 'planes above/below X μm'.

Line 620: "radius of 100 μm radius" – **AR:** Deleted second 'radius'.

Line 675: The alpha-damage density should also be mentioned. – **AR:** We have added that alpha-damage density is also influencing (U-Th-Sm)/He dates.

Lines 687, 690: "C/Ma" --> C/Myr – **AR:** Changed to 'C/Myr'.

Lines 730, 736: Tipathy --> Tripathy-Lang – **AR:** Corrected to 'Tripathy-Lang'.

Line 797: ablation bit --> ablation pit – **AR:** Corrected to 'pit'.

**AR:** Thanks for the detailed corrections, we will change the manuscript accordingly.

**RC1:** Line 835: "Our observations revealed that radionuclide zoning is not an anomaly but a prevalent occurrence in both apatite and zircon." I am afraid it is already a common knowledge. Maybe it does not fit in this form in the conclusions.

**AR:** We do agree that this is common knowledge, but we provided a detailed analyses of our LA-ICP-MS data on this and think that it is still important to report. Whole grain (U-Th-Sm)/He analyses is usually ignoring this and/or interpreting date/eU relations that can be an artefact of zonation.

**RC1:** Line 840: The issue of the recommended minimum number of grains is important, however, the details of their estimation method is not completely transparent.

**AR:** Thanks for the clarification, to fill this gap in transparency, we will move some of the details from the supplement data to the main text.

**RC1:** Figs. 2, 5, 6: Explain, please, the "Distance" on the X axis. Is it from the center of the grain? Consider a more detailed explanation in the captions – without the text it is difficult the follow.

**AR:** We changed the X axis label to Distance from center/rim in Figs. 2, 5 and 6.

**RC1:** In Fig. 2 (at the comparison of cylindrical and spherical geometries) it should be also indicated that the "distance" is perpendicular to C-axis.

**AR:** Good point, we added this information to the figure caption.

**RC1:** In Fig. 6 the "distance" in the left and right panels are different.

**AR:** To be consistent we change the X axis in panels A to D to start with 0 from the center of the grain.

**RC1:** Fig. 4: (A): add a color scale. (B): To be consequent, and use blue line for the U concentration. Delete Th and Sm from the caption, when the vertical axis indicates only U.

**AR:** Thanks for the details, we changed the figure accordingly.

**RC1:** SD1: It is not clear, how the laser ablation ICPMS measurements were done. Did the authors use (i) down-hole (drilling in a spot) analyses or (ii) the zonation is detected across polished grains by LA line analyses. In case of (i) how was considered/compensated the down-hole fractionation?

**AR:** The data was generated by drilling in a spot and correction for down-hole fractionation was done based on NIST612/610 calibration standards using 43Ca (for apatite) and 29Si (for zircon). We added the required details to the supplement (SD1).

**RC1:** SD3: It is not clear how the radionuclide heterogeneity was considered at the calculation. Did the authors take the variation detected in the own laboratory or took synthetic, modelled zonation patterns?

**AR:** We have used the radionuclide measurements of our own laboratory. We will add this information in the revised manuscript.

**RC1:** Fig. S8: Correct "Fig. X", please.

**AR:** We replaced all occurrences of 'Fig. X' in the supplement with the correct reference to the supplement figures.

**RC1:** The References needs several corrections

Missing:

Brown et al., 2013 – **AR:** Added.

Tripahy-Lang et al., 2013 – **AR:** Added.

Boyce et al., 2006 – **AR:** Added.

Evans et al., 2015 – **AR:** Added.

Ketcham et al., 2011 – **AR:** Added.

Wolf et al., 1998 – **AR:** Added.

Horne et al., 2018 – **AR:** Changed to Horne et al., 2019.

Vermeesch et al., 2012 – **AR:** Added.

Dunkl et al., 2024 – **AR:** Added.

Flowers and Farley, 2012 – **AR:** Added.

Does not cited in the text:

McDowell et al., 2005 – **AR:** Reference deleted.

Alphabetic order:

Horne / House – **AR:** Order corrected.

**RC2:** You might consider revising the title to better convey the manuscript's content. Based on the title I had assumed that this paper would present laser-ablation (U-Th)/He data and interpret it, but no laser-ablation (U-Th)/He data are presented or interpreted and except the eU zonation data this is entirely a modeling study.

**AR:** We do agree, that the title might be somewhat misleading, and the reader might expect to see observational data. To make clear that most of the shown data is generated by models, we added 'modelling perspective' to the title.

**RC2:** It may be helpful to include a first figure to be referred to in the introduction that is a schematic showing why a whole grain date with an alpha-ejected rim will be younger than an in situ date in the core of the grain. This would be effective in explaining the basic concepts to the reader before getting into the details in the rest of the manuscript.

 **AR:** We followed this suggestion and prepared a schematic figure showing the basic difference between whole grain and in-situ (U-Th-Sm)/He method and the resulting date (new Fig. 1). In addition we added a small section explaining the basic difference between whole grain and in-situ date while referring to the figure.

**RC2:** On several of the figures it would be helpful to show the date profiles across the grains to show how the He concentrations relate to the laser-ablation (U-Th)/He dates. For example, on Figures 2, 10, and 11.

**AR:** OK, we agree that this information might be helpful. We have calculated in-situ dates for all corresponding experiments and updated the corresponding figures and captions (see Fig. 3, 11, 12, Fig. S9).

**RC2:** Infinite cylinder vs. sphere geometry: In Figure 2, it appears that the dates in each grain's core could be substantially different for the sphere and infinite cylinder models - is that right or not? It'd be helpful to plot the date profiles as suggested above for this reason. Also, what are the whole grain dates for each of these models and how much do they differ for the sphere vs. infinite cylinder models? This would be worth discussing and then better justifying why an infinite cylinder geometry should be used instead of a sphere (especially if the dates are different between the

two geometries). The choice of geometry appears to matter a lot more than, for example, whether complete stopping distances are used or not as shown in Figure 5.

**AR:** The corresponding whole-grain and in-situ dates have been added to the figure. An infinite cylinder of the same geometric radii as a sphere has a sphere-equivalent radius of 1.5 times the sphere equivalent. Therefore, the in-situ and whole-grain dates are older in the infinite cylinder compared to a sphere with a similar radius. We have added a sentence stating that choosing the right geometry is essential in in-situ dating.

**RC2:** I appreciate the modeling exercises in this paper. Are the spot sizes used in the Figures 7, 10, 11 and 12 models analytically reasonable given the He, U, and Th concentrations of the modeled grains? Given real analytical and blank limits? I think this is likely true for the Figure 10 and 11 examples, but am unsure of the others. The He detection limit on the Tuebingen system and how this relates to the youngest grains that can be measured for given pit volumes are noted in Lines 62-67, but it's unclear if all of the modeling exercises are done for analytically reasonable scenarios. It would be good to explicitly state this for each modeling exercise. Since an important conclusion of the paper is that that in situ (U-Th-Sm)/He dating can improve thermal history interpretations, to make this point I think it's important to be clear that the theoretical data modeled are analytically feasible.

**AR:** This is an important point and we will address this in the revised manuscript. There is a trade-off between the spot size (or better volume) required to achieve a reasonable analytical uncertainty with the in-situ method. What we will do in the revised manuscript, is to report which spot size results in an analytical uncertainty above for instance 10%, likely by reporting those scenarios with a different colour. We can use our lab-specific values for reference, but this will be lab-specific and dependent on the reproducibility of the line-blank He content.

**RC2:** Given your detailed discussion of the importance of radionuclide zonation effects, including a thermal history modeling exercise (like those in Figures 10 and 11) that includes zonation would be a good addition to this paper. Radionuclide zonation is part of the challenge of laser-ablation (U-Th)/He dating because of alpha redistribution combined with the inability to measure the exact same volumes for daughter and parent. This would make for a slightly more complex, but arguably more realistic, thermal history modeling example.

**AR:** Since there are numerous possibilities to do this exercise, we concentrated on one of the most unpleasant scenario with a thin higher concentrated rim. We invert the theoretical data using the two in-situ approaches and the whole-grain approach. Results are shown as a new figure and discussed in the text.

**RC2:** Details given in the annotated manuscript pdf.

**AR:** Thanks for the detailed correction of the manuscript. We went through all comments (see comments by the author above) and minor corrections. We do agree with all reviewers corrections and revised the manuscript accordingly.

**AEC:** Could you clarify what you mean with "depth-resolved information"? Most in-situ (U-Th-Sm)/He studies pair a single helium pit with a single LA-ICP-MS pit, where the former tends to be larger than the latter. Using this procedure, my experience is that the effect of compositional zoning is stronger for in-situ data than for whole-grain analyses, not weaker (Vermeesch et al., 2023; doi:10.5194/gchron-5-323-2023). This is because, for in-situ dating, the radioactive parent (U,Th,Sm) and radiogenic daughter elements are sourced from and measured in separate volumes. In contrast, whole-grain degassing and dissolution ensures that the parents and daughter are sourced from and measured in the same volume.
The consequences of this geometry are nicely illustrated by Figure 4 of your manuscript. For this simulated crystal, the whole-grain date may be 10% wrong (Fig 8), but the in-situ date could be off by 50%. My experience is that this is not an unusual situation.

**AR:** We do agree that in the case of radionuclide zoning the in-situ (U-Th-Sm)/He approach won't be the ,ideal' method and will likely produce biased data. Zoning, however, can be recognized and those grains rejected from further analyses. We added a few sentences to make this point clear.

**AEC:** Furthermore, most of your examples assume that U-Th-Sm-zoning is smooth, whereas in reality it can be highly irregular. It is well known that compositional zoning is scale invariant (fractal), so smaller spot sizes do not improve the reproducibility and representativity of the measurements.
In summary, I am not sure why your paper maintains that in-situ dates are more accurate than whole-grain dates. I think that they are often less accurate, at least using the 'conventional' approach in which a single helium pit is combined with a single LA-ICP-MS pit. Of course, the problem could be greatly reduced using Danisik et al. (2017, doi:10.1126/sciadv.1601121)'s mapping approach. However, I'm not sure if this technique is practical on a routine basis yet (otherwise it would be more

widely used by now). But perhaps it is the only way to address the zoning issue, albeit under the tenuous assumption of 3D symmetry for 2D zoning patterns.

**AR:** In case grains are zoned, I do agree. The in-situ method, however, has the potential to identify zoned grains and does take away the uncertainty we always had in whole-grain dating with zoning. You probably also experienced dating 4-5 grains and getting either a large spread in single grain dates or dates that are rather reproducible, but incompatible with other thermochronometers. In such cases it would be helpful to have known the radionuclide distribution and I am optimistic that the in-situ method will help identify such problematic grains/samples. We went through the text and added the above information.

**AEC:** I am not asking you to drastically expand your paper with additional case studies. However, I would encourage you express your thoughts on this issue more clearly in a suitably revised version of your manuscript.

**AR:** We went through the manuscript and added information/thoughts on the expected accuracy of in-situ dates.

**AEC:** I inspected the Matlab script in your Zenodo repository and noticed a few redundant initialisations that you may want to clean up. This comment aside, I would encourage you to share some more of the code that you used to generate the figures in your paper. This will likely be useful to readers who want to apply your methods to their own work.

**AR:** We used Valgrind to automatically detect issues with the c++ file. There is only one issue with the 'dvector' initialization. Since 'dvector' is used quite often in the code, we could not identify the issue, but it is causing only 28 kb of unused space when calling the c++ file (not too much to spend more time on finding the issue). We also uploaded to Zenodo some Matlab scripts to plot He profiles and in-situ dates such as those in Fig. 8 of the manuscript.